# Asthma reduces glioma formation by T cell decorin-mediated inhibition of microglia

Jit Chatterjee[1], Shilpa Sanapala[1], Olivia Cobb[1], Alice Bewley[1], Andrea K. Goldstein[1], Elizabeth Cordell[1], Xia Ge[2], Joel R. Garbow[2], Michael J. Holtzman [3] & David H. Gutmann [1]✉

To elucidate the mechanisms underlying the reduced incidence of brain tumors in children with Neurofibromatosis type 1 (NF1) and asthma, we leverage *Nf1* optic pathway glioma (*Nf1*[OPG]) mice, human and mouse RNAseq data, and two different experimental asthma models. Following ovalbumin or house dust mite asthma induction at 4–6 weeks of age (WOA), *Nf1*[OPG] mouse optic nerve volumes and proliferation are decreased at 12 and 24 WOA, indicating no tumor development. This inhibition is accompanied by reduced expression of the microglia-produced optic glioma mitogen, Ccl5. Human and murine T cell transcriptome analyses reveal that inhibition of microglia Ccl5 production results from increased T cell expression of decorin, which blocks Ccl4-mediated microglia Ccl5 expression through reduced microglia NFκB signaling. Decorin or NFκB inhibitor treatment of *Nf1*[OPG] mice at 4–6 WOA inhibits tumor formation at 12 WOA, thus establishing a potential mechanistic etiology for the attenuated glioma incidence observed in children with asthma.

---

[1] Department of Neurology, Washington University School of Medicine, St. Louis, MO, USA. [2] Mallinckrodt Institute of Radiology, Washington University School of Medicine, St. Louis, MO, USA. [3] Department of Pulmonary and Critical Care Medicine, Washington University School of Medicine, St. Louis, MO, USA. ✉email: gutmannd@wustl.edu

Prior case-controlled clinical studies have revealed a reduced incidence of brain tumors in children with T-cell-mediated disorders, including asthma[1–4]. Similarly, in the Neurofibromatosis type 1 (NF1) brain cancer predisposition, we previously reported an inverse association between asthma and optic gliomas, the most common brain tumor in this population of at-risk children[5]. To study this inverse association between asthma and brain tumor incidence in a cancer predisposition syndrome in which atopic conditions reduce glioma incidence, we leveraged a genetically engineered mouse model of NF1-low-grade glioma of the optic pathway. This authenticated preclinical murine model exhibits high tumor penetrance (>95%) and a well-characterized temporal course of glioma progression[6,7]. As such, $Nf1^{flox/mut}$, GFAP-Cre ($Nf1^{OPG}$) mice develop low-grade gliomas of the optic nerves and chiasm by 12 weeks of age, with accompanying increases in optic nerve volume and proliferation, as well as microglia and T-cell infiltration, beginning after 6 weeks of age[8,9]. Moreover, the tumor-associated microglia express high levels of Ccl5, a growth factor both necessary and sufficient to drive $Nf1$-optic glioma formation and continued growth[10,11]. Germane to the pathobiology of inflammatory disease, infiltrating T cells in the optic nerve produce Ccl4, which in turn stimulates microglia to produce Ccl5 through activation of the NFκB signaling pathway[12]. Moreover, T cell entry and activation are required for $Nf1$-optic glioma growth in mice, and murine optic glioma stem cells do not form tumors in T cell-deficient mice[13]. As such, $Nf1^{OPG}$ mice provide a tractable genetic platform to elucidate the molecular and cellular mechanisms responsible for the reduced incidence of brain tumors in children with a T cell-mediated disease like asthma. Herein, we establish a mechanistic link between asthma and gliomagenesis by demonstrating that asthma induces decorin expression in T cells to reduce microglia support of $Nf1$ optic glioma growth.

## Results

**Asthma induction inhibits $Nf1^{OPG}$ formation.** To determine whether experimental asthma induction reduces murine glioma formation and growth in vivo, $Nf1^{OPG}$ mice were initially treated with ovalbumin (OVA) or house dust mite (HDM) between 4 and 6 weeks of age and euthanized at 12 weeks of age when optic gliomas are evident in >95% of mice (Fig. 1a). Experimental asthma induction was confirmed by H&E staining of lung tissue, demonstrating infiltration of immune cells in peribronchiolar locations, consistent with submucosal inflammation after OVA or HDM challenge conditions compared to PBS controls (Supplementary Fig. 1a). In addition, there were PAS$^+$ airway epithelial cells with morphology consistent with goblet cells, as well as increased serum IgE in OVA-and HDM-treated mice (Supplementary Fig. 1b, c). At 12 weeks of age, the optic nerves of OVA- and HDM-treated $Nf1^{OPG}$ mice exhibited reduced volumes (Fig. 1b) and proliferation (%Ki67$^+$ cells; Fig. 1c), without changes in microglia (%Iba1$^+$ cells, Fig. 1c) or T cell (CD3$^+$ cells/nerve) (Fig. 1c) content. Importantly, this decrease in optic nerve volume and proliferation was also observed at 24 weeks (18 weeks after asthma induction) (Fig. 1d–f), demonstrating a durable effect on gliomagenesis.

Interestingly, optic nerve volumes and proliferation did not return to wild-type (WT) levels following asthma induction. Since $Nf1^{OPG}$ mice, like patients with NF1-OPG, are heterozygous for a germline $Nf1$ gene mutation in every non-neoplastic cell in their bodies, we analyzed the optic nerves of mice with a germline $Nf1$ mutation ($Nf1^{+/-}$ mice). These $Nf1^{+/-}$ mice have increased optic nerve volumes by direct measurements (Supplementary Fig. 2a, b) and increased optic nerve areas by magnetic resonance imaging (Supplementary Fig. 2c), similar to some patients with NF1

(Supplementary Fig. 2d)[14]. In addition, $Nf1^{+/-}$ mouse optic nerves had increased microglia content (%Iba1$^+$ cells; Supplementary Fig. 2e), but no change in proliferation (%Ki67$^+$ cells; Supplementary Fig. 2f) or T cell content (CD3$^+$ cells; Supplementary Fig. 2g) relative to WT mice, thus providing a clear separation of mice with optic gliomas (Supplementary Fig. 2h, i). In this regard, OVA- and HDM-mediated asthma induction reduces optic nerve volumes and proliferation to levels comparable to $Nf1^{+/-}$ mice (Supplementary Fig. 2j, k), thus establishing that asthma blocks optic glioma formation.

**Asthma reduces $Nf1^{OPG}$ Ccl5, but not Ccl4, levels.** Prior studies using $Nf1^{OPG}$ mice revealed that optic glioma progression and maintenance requires T cells and microglia. In this manner, T cells are activated by $Nf1$-mutant neurons through the production of Ccl4 that stimulates microglia to produce Ccl5 (Supplementary Fig. 3a), the critical growth factor required for $Nf1$-optic glioma formation and continued growth[10–12]. To establish where T cells are activated to express Ccl4, we analyzed $Ccl4$ RNA expression in the blood, optic nerves, and draining cervical lymph nodes of $Nf1^{OPG}$ mice (Fig. 2a, b). Consistent with our previous findings that $Nf1^{+/-}$ neurons (retinal ganglion cells) stimulate T cells to produce Ccl4, the highest levels of Ccl4 were observed in the optic nerves (Fig. 2b).

Next, to determine whether experimental asthma attenuates the increase in Ccl4 and Ccl5 expression observed in the optic nerves of $Nf1^{OPG}$ mice at 12 (Fig. 2c and Supplementary Fig. 3b, c) and 24 (Supplementary Fig. 3d, e) weeks of age, quantitative real-time PCR was performed. No change in $Ccl4$ expression at either 12 (Fig. 2d) or 24 (Supplementary Fig. 3f) weeks of age were observed, indicating that OVA and HDM treatment does not affect T cell Ccl4 production. In striking contrast, $Ccl5$ expression was reduced in the optic nerves of $Nf1^{OPG}$ mice following OVA and HDM treatment at 12 (Fig. 2e) and 24 (Supplementary Fig. 3g) weeks of age. As expected, Ccl4 levels were increased in the draining cervical lymph nodes of $Nf1^{OPG}$ mice relative to WT controls (Fig. 2f); however, Ccl4 expression was unchanged following OVA and HDM exposure (Fig. 2g). Importantly, OVA or HDM treatment does not induce a switch between conventional microglia phenotypes (M1 and M2) (Supplementary Fig. 3h–l). The findings reveal that experimental asthma induction impairs microglia Ccl5 support of optic glioma formation and growth without altering T-cell Ccl4 expression.

**Asthma reduces T cell induction of Ccl5 in microglia.** T cell Ccl4 production in $Nf1^{OPG}$ mice results from either T cell activation (Supplementary Fig. 4a, b) or $Nf1$-mutant neuron production of the pleiotrophin family member, midkine (Mdk), in the setting of $Nf1$-OPG (Fig. 3a)[12]. Consistent with unchanged T cell Ccl4 levels following experimental asthma induction, $Mdk$ levels were similar in OVA- and HDM-treated $Nf1^{OPG}$ mice relative to controls (Fig. 3b), and Mdk does not induce Ccl5 production in microglia (Supplementary Fig. 4c). In addition, naive T cells from PBS-treated (controls) and OVA-treated $Nf1^{OPG}$ mice responded similarly to Mdk stimulation, producing equivalent levels of Ccl4 (Fig. 3c).

Since there were no changes in MDK expression, we then sought to determine whether the impaired microglia Ccl5 production resulted from other potential changes in the immune microenvironment of $Nf1$-optic gliomas. First, it is conceivable that experimental asthma induction alters T cell population infiltration into the optic nerves. Previously, we demonstrated that CD8$^+$ T cells predominate in $Nf1$-optic gliomas (Supplementary Fig. 4d)[12], which was unaffected in $Nf1^{OPG}$ mice following either OVA or HDM treatment (Fig. 3d, e). In addition,

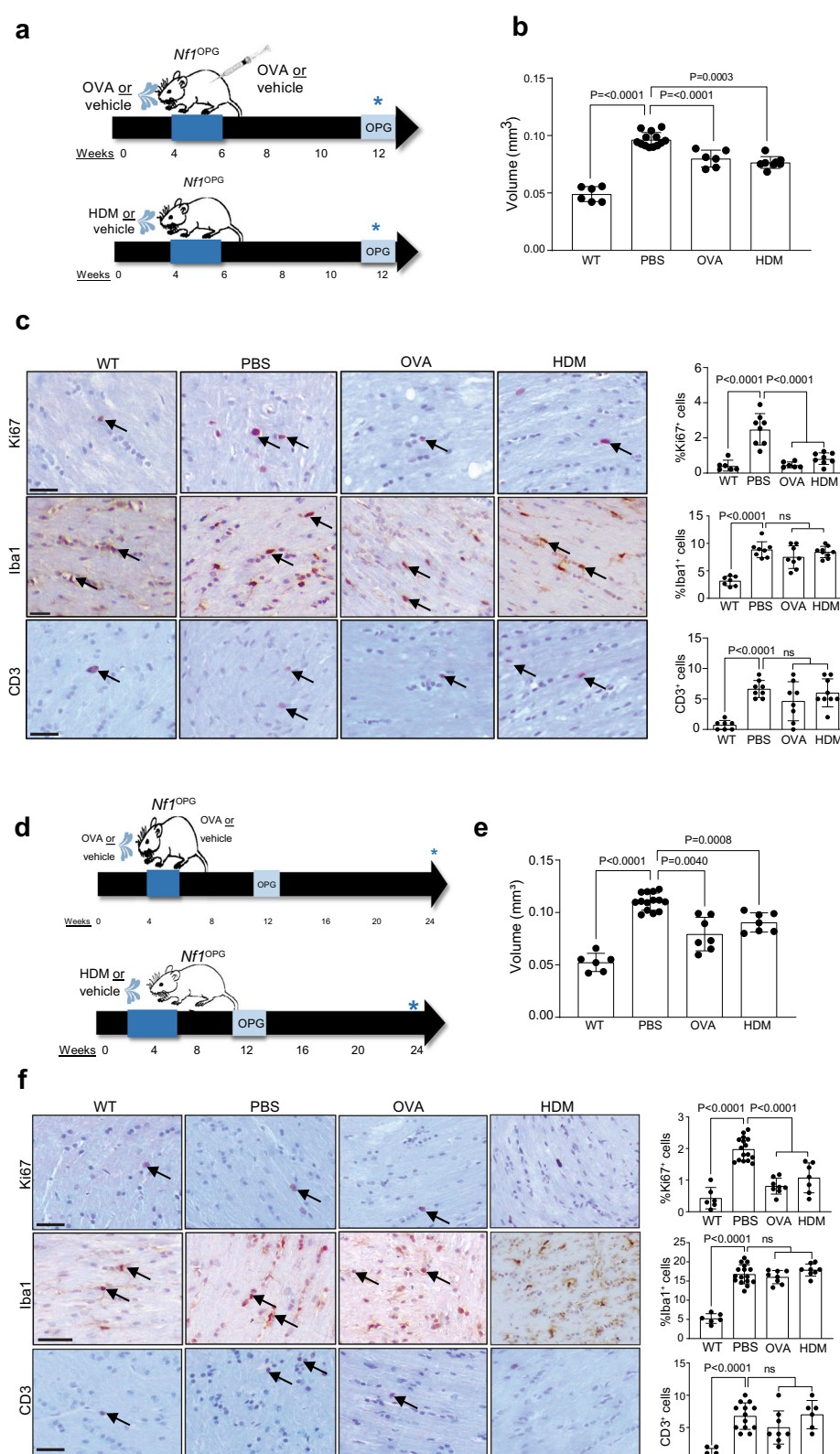

no Foxp3[+] cells (Tregs) were observed in optic gliomas from Nf1-OPG mice treated with PBS, HDM, or OVA (Supplementary Fig. 4e). Second, it is possible that known suppressors of microglia function, like TGFβ[15], or other inflammatory mediators[16–18] induced by asthma could be responsible. Treatment of microglia with either TGFβ alone or in combination with Ccl4 had no effect on Ccl5 expression (Supplementary Fig. 4f). While IL-3, but not IL-4 or IL-5, was induced in T cells following OVA treatment (Supplementary Fig. 4g–i), IL-3 treatment had no effect on either T cell Ccl4 or microglia Ccl5 expression (Supplementary Fig. 4j, k). Third, asthma induction could increase peripheral macrophage infiltration; however, no change in Tmem119[+] cell

**Fig. 1 Asthma induction inhibits $Nf1^{OPG}$ formation. a** Schematic representation of the treatments used to induce asthma in $Nf1^{OPG}$ mice. $Nf1^{OPG}$ mice were treated between 4 and 6 weeks of age with either OVA or HDM, while control $Nf1^{OPG}$ mice received PBS only (vehicle). Isolated optic nerves were analyzed at 12 weeks of age. OVA and HDM treatments reduced (**b**) optic nerve volumes (WT, $n = 6$; PBS, $n = 14$; OVA, $n = 6$; HDM, $n = 8$) and (**c**) percentage of proliferating (%Ki67+) tumor cells within the optic nerves of $Nf1^{OPG}$ mice relative to the vehicle-treated groups (WT, $n = 6$; PBS, $n = 8$; OVA, $n = 8$; HDM, $n = 9$). No difference in microglia (%Iba1+ cells) or T cell (CD3+ cells) content were observed between OVA- or HDM-treated mice and their respective controls. Reduced optic nerve volumes (**e**) and proliferation (%Ki67+ cells; **f**) were also observed at 24 weeks of age after OVA and HDM treatment (**d**) between 4 and 6 weeks of age (WT, $n = 6$; PBS, $n = 13$; OVA, $n = 8$; HDM, $n = 7$). Data are presented as the means ± SEM. **c**, **f** Scale bars, 40 μm. One-way ANOVA with Bonferroni post hoc correction. Exact $P$ values are indicated within each panel (ns, not significant). From left to right in each panel: **b** $P < 0.0001$, $P < 0.0001$, $P < 0.0003$; **c** $P = 0.0001$, ns; **e** $P < 0.0001$, $P = 0.0040$, $P = 0.0008$; **f** $P < 0.0001$, $P < 0.0001$, $P < 0.0001$, ns, $P < 0.0001$, ns.

(microglia) content was observed following OVA or HDM treatment (Supplementary Fig. 4l). Moreover, splenic macrophages fail to induce Ccl5 expression following Ccl4 treatment (Supplementary Fig. 4m). Similarly, OVA and HDM treatments had no effect on other functions of microglia, such as phagocytosis and viability (Supplementary Fig. 4n, o).

To test whether T cells from $Nf1^{OPG}$ mice exposed to OVA or HDM could induce Ccl5 expression in microglia, we incubated naive microglia with antibody-mediated activated T cell-conditioned medium (TCM) from OVA- or HDM-treated $Nf1^{OPG}$ mice. Consistent with the whole tissue results demonstrating reduced Ccl5 expression in the optic nerves of $Nf1^{OPG}$ mice following OVA or HDM treatment, CD3+ TCM from both OVA (Fig. 3f) and HDM (Fig. 3g) mice reduced microglia Ccl5 expression. Moreover, this TCM attenuation of Ccl5 was observed in both CD4+ and CD8+ T cell populations following experimental asthma induction in $Nf1^{OPG}$ mice (Fig. 3h, i).

**Asthma induces T cell decorin expression to inhibit microglia Ccl5 production**. Since T cell Ccl4 levels were not reduced by experimental asthma induction, we hypothesized that OVA- and HDM-treated $Nf1^{OPG}$ mouse T cells might produce an inhibitor of microglia Ccl5 production. To identify this inhibitor, we performed bulk RNA sequencing on isolated spleen CD3+ T cells from $Nf1^{OPG}$ mice treated with PBS (control) or OVA (asthma induction) (Fig. 4a and Supplementary Fig. 5a, b). Following filtering of differentially expressed transcripts ($P$ values ≤ 0.01, false discovery rate ≤ 0.05, and log fold change ≥ 5), only one of the top 20 differentially expressed genes was predicted to be secreted (Fig. 4b). This gene, decorin (Dcn), is a member of the small leucine-rich proteoglycans (SLRPs)[19], originally identified as a collagen-binding protein[20]. Importantly, loss of decorin attenuates asthma in mice[21], and OVA treatment of Dcn-deficient mice results in diminished lung pathology relative to WT mice[22].

As predicted by the RNA sequencing results, Dcn RNA expression was increased in CD3+ T cells from the spleen (Supplementary Fig. 5c–e), cervical lymph nodes (Supplementary Fig. 5f–h), and optic nerve (Supplementary Fig. 5i–k) of OVA- and HDM-treated $Nf1^{OPG}$ mice. Consistent with the ability of both CD4+ and CD8+ TCM to suppress microglia Ccl5 expression, both CD4+ and CD8+ splenic T cells from OVA-treated $Nf1^{OPG}$ mice exhibited elevated Dcn levels relative to control PBS-treated mice (Fig. 4c). Relevant to the human condition, DCN expression was also increased in CD4+ and CD8+ T cells from patients with non-steroid-dependent asthma relative to healthy controls (GSE31773; Fig. 4d).

With similar levels of decorin (Dcn) RNA expression in the blood, optic nerves, and draining cervical lymph nodes of $Nf1^{OPG}$ mice (Fig. 4e), we next wanted to determine whether decorin could directly suppress microglia Ccl5 production. Decorin protein levels were measured in the T cell-conditioned medium (TCM) from OVA-treated $Nf1^{OPG}$ mouse CD3+ T cells relative to their PBS-treated counterparts (Fig. 4f).

Using the concentration (800 pg/ml) of decorin detected in OVA-treated $Nf1^{OPG}$ mouse CD3+ TCM, we found that microglia Ccl5 production was reduced to 400 pg/ml (40% decrease; Fig. 5a)—levels previously shown not to increase $Nf1$ mouse optic glioma stem cell growth in vitro[12]. To further explore the direct effect of decorin on microglia Ccl5 production, we quantified Ccl5 expression in primary microglia in response to Ccl4 exposure over a range of decorin concentrations in vitro (Fig. 5b). In contrast, other members of the SLRP family, specifically biglycan (Bgn), were not induced in OVA-treated $Nf1^{OPG}$ mice relative to PBS controls (Supplementary Fig. 6a), and biglycan did not reduce microglia Ccl5 production (Supplementary Fig. 6b). Taken together, these results demonstrate that T cells from $Nf1^{OPG}$ mice treated with HDM or OVA produce decorin, which impairs the ability of T cells to stimulate microglia, thus interfering with a tumor supportive microenvironment for murine $Nf1$-optic glioma formation.

Since the receptors for Ccl4 (Ccr5 and Ccr8) are both expressed in microglia, we sought to determine which receptor might be responsible for decorin-mediated suppression of Ccl4 induction of microglia Ccl5. While decorin (DCN) treatment reduced microglia Ccl5 production, the addition of a Ccr8 inhibitor (MCV) had no further effect in suppressing microglia Ccl5 production. In contrast, the addition of a Ccr5 inhibitor (AZ058) further reduced activated T cell media-induced microglia Ccl5 levels (Fig. 5c). Based on these results, we conclude that decorin binds to the Ccr8 receptor to reduce microglia Ccl5 production (Fig. 5d).

To demonstrate that decorin is *sufficient* to block optic glioma formation, $Nf1^{OPG}$ mice were treated with decorin (1 mg/ml) from 4 to 6 weeks of age, and then analyzed at 12 weeks of age (Fig. 5e). As observed in the experimental models of asthma (OVA and HDM induction) where decorin expression was induced in T cells, decorin treatment of $Nf1^{OPG}$ mice reduced tumor proliferation (% Ki67+ cells), with no change in optic nerve volumes, microglia (Iba1+ cells), or T cells (CD3+ cells) in vivo (Fig. 5f, g).

**Decorin inhibits microglia Ccl5 production through NFκB inhibition**. Since T cell Ccl4 induction of microglia Ccl5 production is controlled by NFkB activation in microglia[12], we measured NFκB activation (IκBα$^{Serine32}$ phosphorylation) in the optic nerves of $Nf1^{OPG}$ mice relative to their WT counterparts. While NFκB activation was higher in $Nf1^{OPG}$ mouse optic nerves compared to WT controls (Fig. 6a), optic nerves from OVA-treated $Nf1^{OPG}$ mice had reduced NFκB activation in vivo (Fig. 6b). Consistent with decorin suppression of $Nf1^{OPG}$ microglia NFκB activation, the addition of decorin (800 pg/ml) reduced microglia NFκB activation (Fig. 6c), p65-NFκB activation (Ser$^{536}$ phosphorylation; Fig. 6d), and p65-NFκB nuclear location (Fig. 6e, f) following activated T cell-conditioned medium (TCM) exposure in vitro. Similarly, both decorin (Supplementary Fig. 6c) and NFkB inhibition (CAPE; Supplementary Fig. 6d) blocked Ccl4 induction of microglia Ccl5 expression in vitro. In addition,

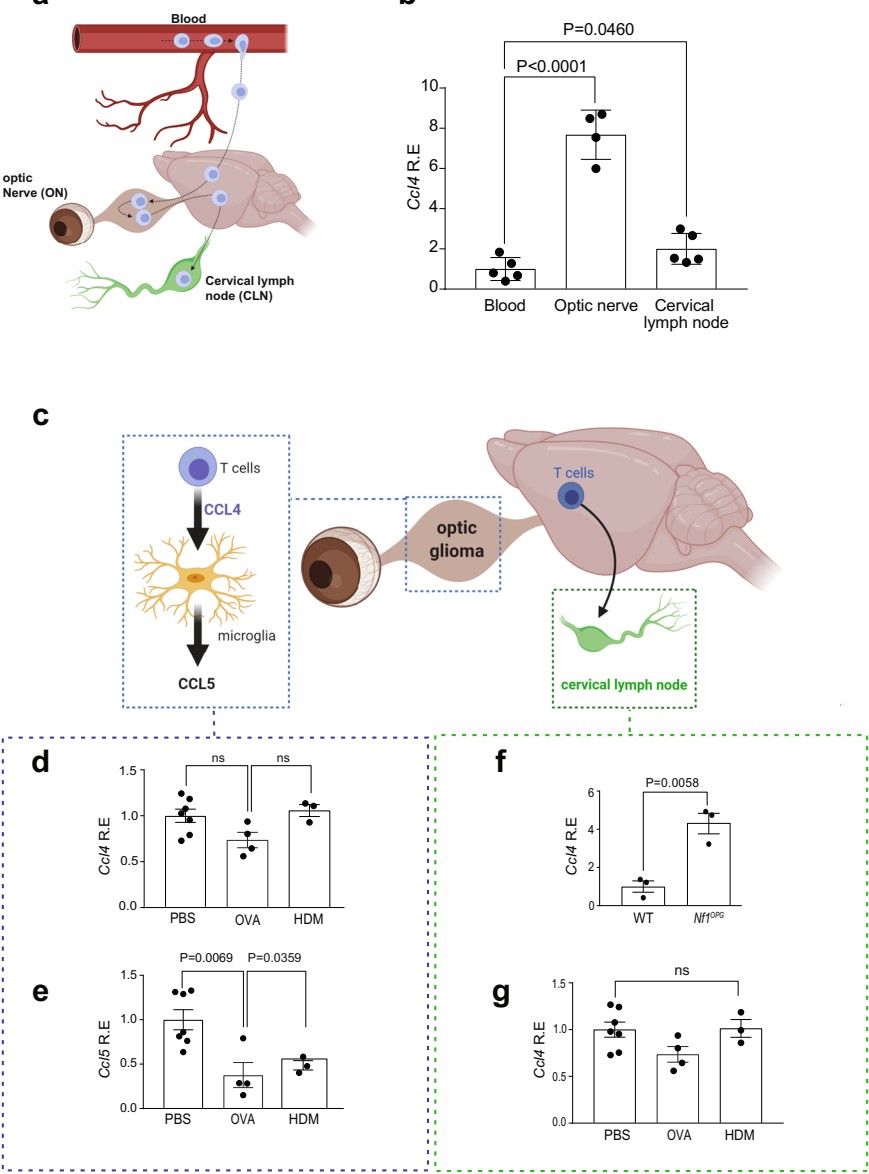

**Fig. 2 Asthma reduces *Nf1*<sup>OPG</sup> Ccl5, but not Ccl4, levels. a** Schematic representation of the journey of T cells from the periphery (blood) into the optic nerve and draining through the cervical lymph node. **b** Increased *Ccl4* mRNA expression was observed in optic nerves ($n = 4$) of *Nf1*<sup>OPG</sup> mice relative to blood ($n = 5$) and draining cervical lymph nodes ($n = 5$). One-way ANOVA with Bonferroni post hoc correction. **c** Schematic representation of the interactions between T cells and microglia in the optic nerve and draining cervical lymph node. **d** No change in *Ccl4* mRNA expression was found in the optic nerves of *Nf1*<sup>OPG</sup> mice treated with OVA ($n = 4$) or HDM ($n = 3$) relative to vehicle-treated ($n = 7$) mice. One-way ANOVA with Bonferroni post hoc correction. **e** Decreased optic nerve *Ccl5* mRNA expression was observed in both OVA- ($n = 4$) and HDM- ($n = 3$) treated mice relative to PBS-treated controls ($n = 7$). One-way ANOVA with Bonferroni post hoc correction. **f** While *Ccl4* gene expression was increased in the cervical lymph nodes of *Nf1*<sup>OPG</sup> mice relative to controls (WT, $n = 3$; *Nf1*<sup>OPG</sup>, $n = 3$), Two-tailed Student's *t* test. (**g**) No change in cervical lymph node *Ccl4* mRNA expression was observed between OVA- or HDM-treated mice (OVA, $n = 4$; HDM, $n = 3$) relative to controls ($n = 7$). One-way ANOVA with Bonferroni post hoc correction. Data are presented as the means ± SEM. Exact *P* values are indicated within each panel (ns, not significant). From left to right in each panel: **b** $P < 0.0001$, $P = 0.0460$; **d** ns, ns; **e** $P = 0.0069$, $P = 0.0359$; **f** $P = 0.0058$; **g** ns.

NFkB inhibition has no effect on microglia viability (Supplementary Fig. 6e).

Lastly, to directly demonstrate that NFκB activation is *necessary* for *Nf1*-optic glioma formation, *Nf1*<sup>OPG</sup> mice were treated with a commercial NFκB inhibitor (10 mg/kg CAPE) from 4 to 6 weeks of age, and analyzed at 12 weeks of age (Fig. 6g). Similar to both decorin-treated and OVA/HDM-induced *Nf1*<sup>OPG</sup> mice, NFκB inhibitor (CAPE) treatment blocked tumor formation (optic nerve volumes, Fig. 6h; tumor proliferation, %Ki67<sup>+</sup>

cells, Fig. 6i). To demonstrate that decorin suppresses T-cell-mediated microglia production of CCL5 by inhibiting NFκB activation, we show that NFκB inhibition reduces *Nf1*<sup>OPG</sup> *Ccl5* expression (Fig. 6j). However, in contrast to the decorin results, NFκB inhibitor treatment also reduced microglia (%Iba1<sup>+</sup> cells) and T-cell (CD3<sup>+</sup>) content within the optic nerves of *Nf1*<sup>OPG</sup> mice relative to the vehicle-treated groups (Fig. 6i), likely reflecting the effect of NFκB inhibition on microglia and T cell migration[23,24].

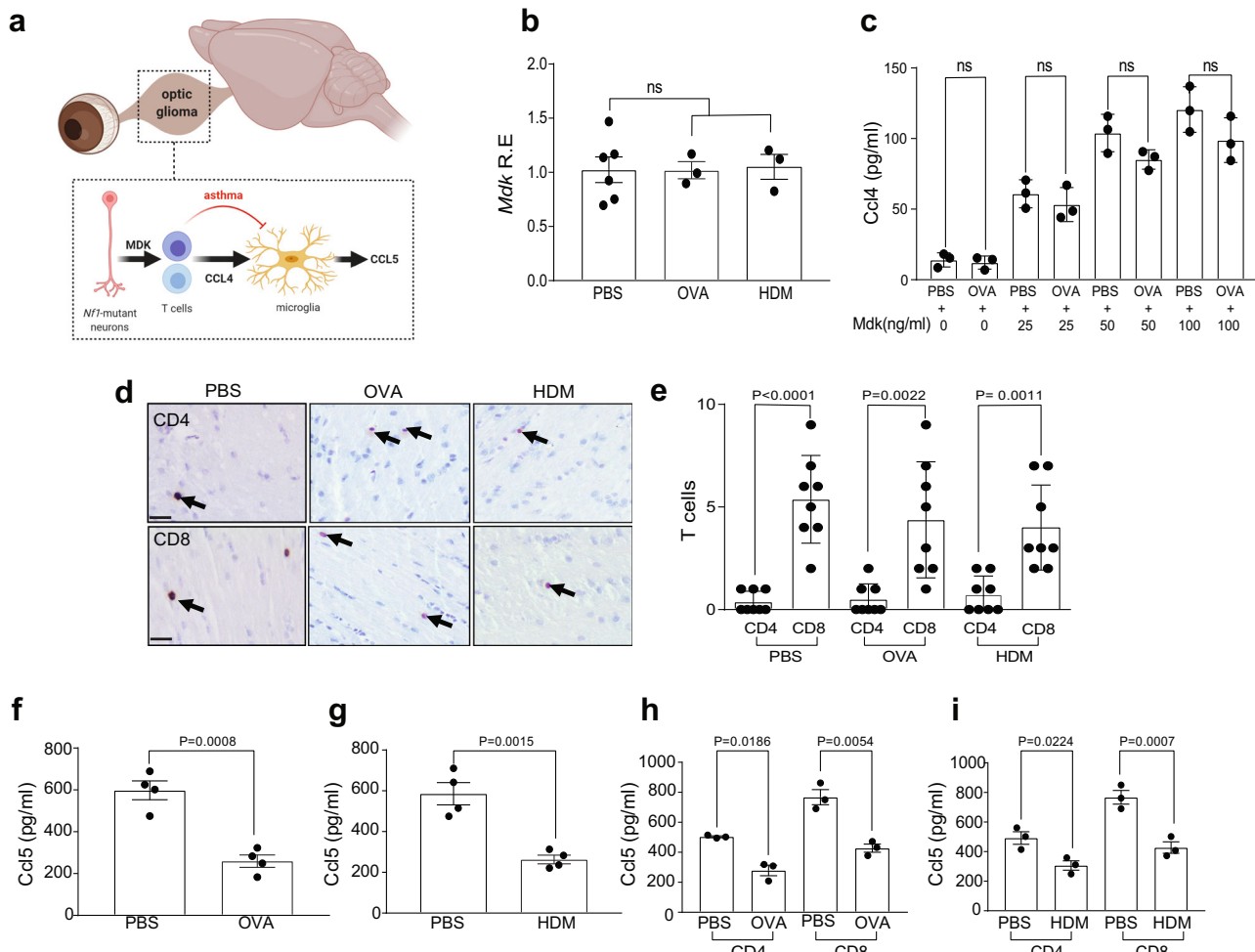

**Fig. 3 Asthma reduces T cell induction of Ccl5 in microglia. a** Schematic representation of the mouse *Nf1*^OPG neuron-immune-cancer cell axis in the setting of asthma. **b** No difference in optic nerve *Mdk* RNA expression was observed between OVA- ($n = 3$) or HDM- ($n = 3$) treated mice and controls ($n = 6$). Bar graphs represent the means ± SEM of independent biological samples. One-way ANOVA with Bonferroni post hoc correction. **c** Ccl4 levels are similarly induced with increasing midkine concentrations (0–100 ng/ml) in CD3+ T cells from OVA- and PBS-treated mice. Data are presented as the means ± SEM of $n = 3$ independent biological samples. One-way ANOVA with Bonferroni post hoc correction. Data are presented as the means ± SEM. **d, e** Similarly increased CD8+, relative to CD4+, T cell content was detected in the optic nerves of *Nf1*^OPG mice treated with PBS ($n = 8$), OVA ($n = 8$) or HDM ($n = 8$). One-way ANOVA with Bonferroni post hoc correction. Activated T cell medium from (**f**) OVA- ($n = 4$) and **g** HDM- ($n = 4$) treated mice induced lower levels of microglial Ccl5 relative to PBS-treated *Nf1*^OPG mice. Two-tailed Student's *t* test. Similar reductions of Ccl5 expression were observed in CD4+ and CD8+ T cells from (**h**) OVA- ($n = 3$) and (**i**) HDM- ($n = 3$) treated mice relative to PBS-treated controls ($n = 3$). One-way ANOVA with Bonferroni post hoc correction. Data are presented as the means ± SEM. **d** Scale bars, 40 μm. From left to right in each panel: **b** ns; **c** ns; ns; ns; ns; **e** $P < 0.0001$, $P = 0.0022$, $P = 0.0011$; **f** $P = 0.0008$; **g** $P = 0.0015$; **h** $P = 0.0186$, $P = 0.0054$; **i** $P = 0.0224$, $P = 0.0007$.

## Discussion

Collectively, the findings presented in this report suggest a model in which brain cancer risk is modified by alterations in circulating T cell function (Fig. 6k), specifically by interfering with immune cell (T lymphocyte-microglia) communication critical for the establishment of a microenvironment supportive of glioma growth. In this manner, we provide a mechanistic explanation for the reported epidemiological risk association between asthma and a reduced incidence of brain tumors in children with and without NF1[25–27]. Second, while allergen-induced asthma is typically linked to CD4+ T cell responses[28], the majority of the T cells in the murine *Nf1*^OPG tumors are CD8+ T cells[9,12]. Since CD8+ T cells also have complementary roles in asthma and autoimmune conditions[29–31] and both murine and human CD4+ and CD8+ T cells both exhibit increased decorin expression in the setting of asthma, future studies will be required to define the individual contributions of each T cell population to the suppression of microglia-mediated optic gliomagenesis. Third, the discovery that decorin, an SLRP critical for experimental asthma induction in mice and elevated in patients with asthma, has an opposing role in glioma biology, raises the intriguing possibility that specific molecules important for causing systemic inflammatory and autoimmune diseases have different effects in the setting of brain cell axis function. In asthma, decorin potentiates IFNγ activity[32,33], such that decorin-deficient mice have reduced lung tissue inflammation[21], while in cancer, it has anti-oncogenic properties, including the prevention of metastatic spread and angiogenesis[34–36]. While the mechanism underlying decorin suppression of NFκB activity in response to CCL4 engagement of its cognate G protein-coupled receptor, CCR5[12] remains to be elucidated, decorin functions in other cell types to block receptor tyrosine kinase (insulin-like growth factor receptor-1, c-MET, and epidermal growth factor receptor) signaling[37–39]. Lastly, the findings described in this report add to a growing literature implicating T lymphocyte in brain function, including learning[40,41] social behavior[42,43] and neurodegeneration in the

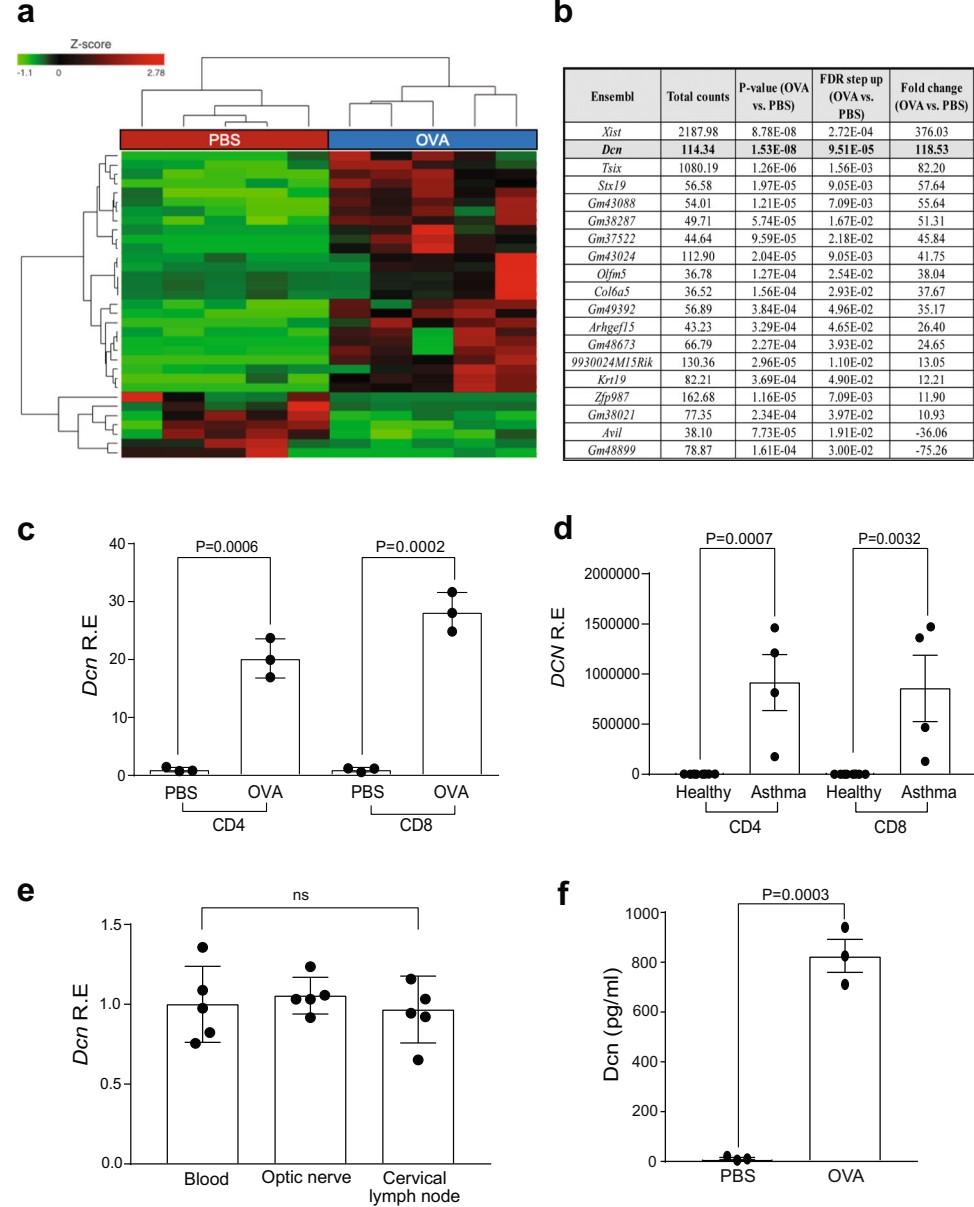

**Fig. 4 Asthma-induced T cell decorin inhibits microglia Ccl5 level. a** Heatmap showing differential gene expression in CD3+ T cells from PBS- versus OVA-treated *Nf1*OPG mice (*n* = 5). **b** Top transcripts increased in T cells from OVA-, relative to PBS-treated, *Nf1*OPG mice calculated with DESeq2 were filtered for only those genes with *P* values ≤ 0.01, false discovery rates ≤ 0.05, and log fold changes greater or equal to ±5. *P* values, false discovery rate (FDR), and fold changes are shown in a tabular form. **c** Increased *Dcn* gene expression was detected in both splenic CD4+ and CD8+ T cells from OVA-treated *Nf1*OPG mice relative to PBS-treated controls. Data are presented as the means ± SEM of *n* = 3 independent biological samples. **d** *DCN* RNA expression was increased in CD4+ and CD8+ T cells from patients with non-steroid-dependent asthma (*n* = 4) relative to healthy controls (*n* = 6; GSE31773). Two-tailed Student's *t* test. Exact p values are indicated within each panel. **e** Similar levels of *Dcn* mRNA expression were observed in T cells from the blood (*n* = 5), optic nerves (*n* = 5), and cervical lymph nodes (*n* = 5) in *Nf1*OPG mice treated with OVA. Data are presented as the means ± SEM of *n* = 5 independent biological samples. One-way ANOVA with Bonferroni post hoc correction. **f** Increased Dcn levels were detected in CD3+ T cell-conditioned medium (TCM) from OVA-treated *Nf1*OPG mice relative to PBS-treated controls by ELISA. Two-tailed Student's *t* test. Exact *P* values are indicated within the panel. Data are presented as the means ± SEM of *n* = 3 independent biological samples. From left to right in each panel: **c** *P* = 0.0006, *P* = 0.0002; **d** *P* = 0.0007, *P* = 0.0032; **e** ns; **f** *P* = 0.0003.

setting of neuronal injury[44,45]. In this manner, it is conceivable that other systemic conditions involving T lymphocytes, such as eczema[26,46,47], rheumatoid arthritis[48,49], and diabetes[50,51], may similarly impact gliomagenesis through overlapping or related mechanisms. Defining the molecular etiologies underlying how these T cell-mediated systemic conditions influence brain function and nervous system disease pathogenesis may also reveal

unique immunomodulatory treatment strategies for neurological disorders.

## Methods
**Mice**. *Nf1*flox/mut; GFAP-Cre (*Nf1*OPG) mice (*Nf1*+/− mice with somatic *Nf1* gene inactivation in neuroglial progenitors at E15.5), *Nf1*+/− mice (neomycin sequence insertion within exon 31 of the murine *Nf1* gene), and *Nf1*flox/flox mice (controls)[6,12]

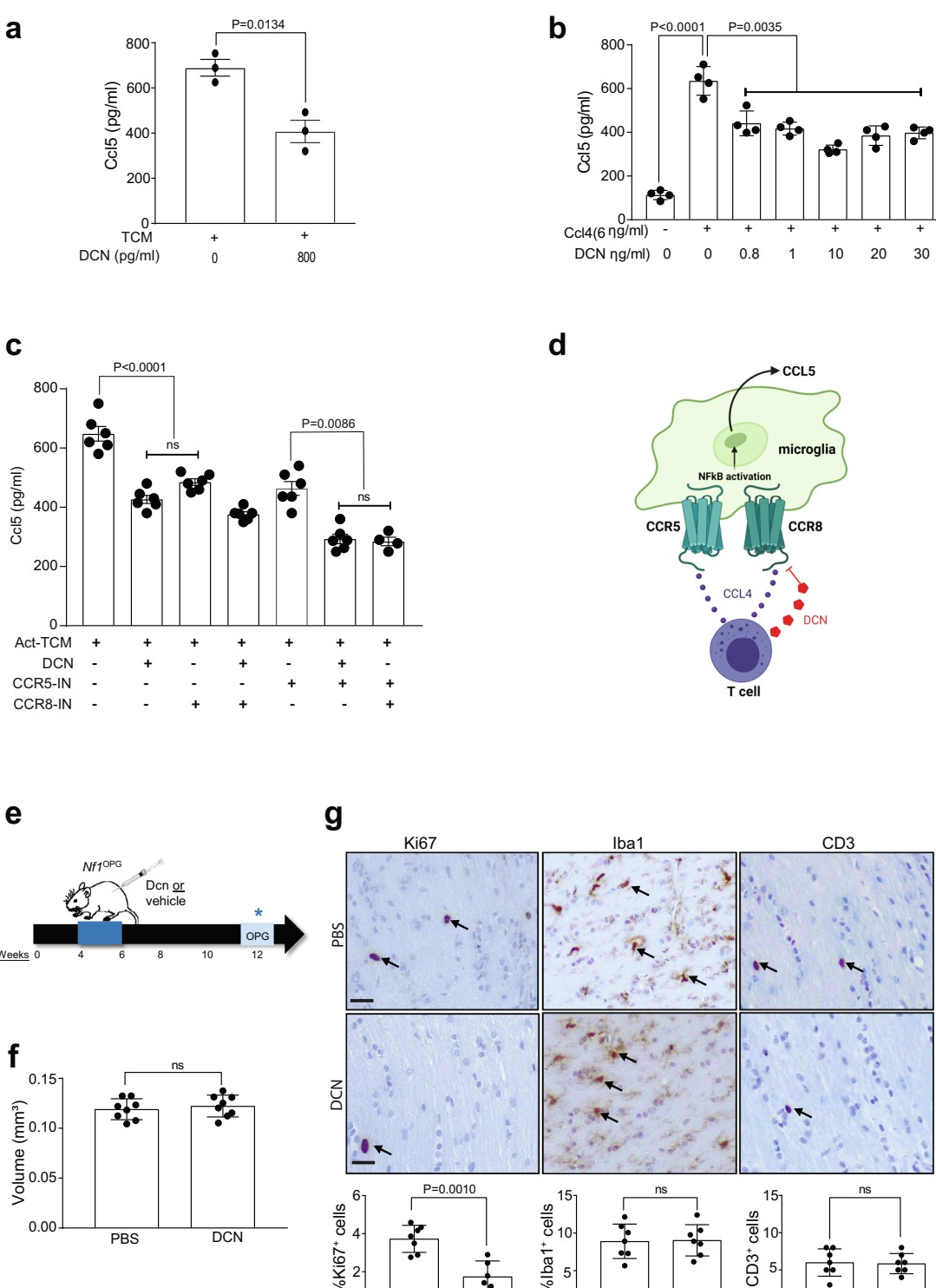

were used in accordance with an approved Animal Studies Committee protocol at Washington University in St. Louis. All mice used in these experiments were maintained on a strict C57BL/6 background and were housed in a specific pathogen-free barrier facility with controlled light-dark cycles (12:12 h) and ad libitum access to food and water.

**Experimental mouse asthma models.** Two different allergen challenges were used to induce experimental asthma in $Nf1^{OPG}$ mice. For the ovalbumin (OVA)-induced asthma model, mice were intraperitoneally injected with 0.5 mL of chicken egg ovalbumin (16 µg/mL) in aluminum hydroxide (2 mg/mL) weekly for two weeks (days −14 and −7), as previously reported[52]. On day 0, mice were anesthetized with an intraperitoneal injection of ketamine (100 mg/kg) and xylazine (10 mg/kg). After a desired level of anesthesia was achieved and verified by loss of the toe pinch reflex, 30 µl of ovalbumin in PBS was administered intranasally. Mice received a total of four intranasal doses of ovalbumin (2 doses, 12 h apart on day 0; 1 dose on day 1; and 1 dose on day 2). For the house dust mite (HDM)-induced asthma model[53,54], mice were anesthetized as described above and one dose of 1 µg HDM in PBS was given intratracheally (day 0). From days 6 to 10, mice were anesthetized and 40 µl of HDM solution (10 µg/ml) in PBS was administered intranasally. Optic nerve and lungs were harvested at 12, 24 weeks of age. Lungs were analyzed for asthma induction, while optic nerves were used for Iba1, Ki67, CD3, CD4, and CD8 analysis by counting the number of immunoreactive cells by immunohistochemistry.

**Fig. 5 Decorin inhibits *Nf1*-OPG formation. a** Decorin (800 pg/ml) attenuated activated TCM (act-Tm)-mediated microglia Ccl5 production ($n = 3$). Two-tailed Student's *t* test. Exact *P* values are indicated within each panel (ns, not significant). **b** Decorin reduces microglia Ccl5 production in response to Ccl4 exposure (6 ng/ml) over a physiologic dose range in vitro ($n = 4$). One-way ANOVA with Bonferroni post hoc correction. Data are presented as the means ± SEM. **c** TCM (activated T cell-conditioned media) induced microglial Ccl5 production was significantly attenuated by decorin. The addition of Dcn with CCR5 has the greatest inhibition similar to the combination of CCR5 and CCR8 inhibitors ($n = 6$). One-way ANOVA with Bonferroni post hoc correction. Data are presented as the means ± SEM. **d** Schematic representation of decorin binding to the CCR8 receptor on microglia, culminating is reduced microglial Ccl5 production by inhibiting NFκB activation. **e** Schematic representation of decorin treatment of *Nf1*[OPG] mice. *Nf1*[OPG] mice were treated between 4 and 6 weeks of age with decorin, while control *Nf1*[OPG] mice received PBS only. Isolated optic nerves were analyzed at 12 weeks of age. **f** Decorin treatment has no change on optic nerve volume, but (**g**) decreased proliferation (%Ki67[+] cells) of *Nf1*[OPG] mice relative to the vehicle-treated *Nf1*[OPG] controls (PBS, $n = 8$, Decorin, $n = 8$). No difference in microglia (%Iba1[+] cells) or T cell (CD3[+] cells) content was observed between decorin-treated mice and their respective controls. Two-tailed Student's *t* test (ns, not significant). **g** Scale bars, 40 μm. From left to right in each panel: **a** $P = 0.0134$, **b** $P < 0.0001$, $P = 0.0035$; **c** $P < 0.0001$, ns, $P = 0.0086$, ns; **f** ns; **g** $P = 0.0010$, ns, ns.

**Mouse treatments**. Four-week-old *Nf1*[OPG] mice were treated with 10 mg/kg NFκB inhibitor (NFκB-IN; Caffeic acid phenethyl ester, Fisher Scientific, 274310) or 1 mg/ml Decorin (R&D System 1060-DE-100) intraperitoneally every other day for 2 weeks between 4 and 6 weeks of age. Brains and optic nerves were harvested when mice reached 12 weeks of age, and Iba1, Ki67, and CD3-positive cells were analyzed by counting the number of cells showing a positive signal after immunohistochemistry.

**Optic nerve volume analysis**. Micro-dissected optic nerves were photographed using Leica DFC 3000 G camera. Measurements were performed using Image-J software beginning at the chiasm (D0) and at 150 (D150), 300 (D300) and 450 μm (D450) anterior to the chiasm to generate optic nerve volume estimates, as previously reported[7]. The volumes for regions 1, 2, and 3 calculated at the three 150 μm high truncated cones were combined using the diameter (D0, D150, D300, and D450) values from each optic nerve measurement. The following equation was used to calculate the total optic nerve volume:

$$V1 = \frac{1}{12}\pi h(D_0^2 + D_0 D_{150} + D_{150}^2)$$

**Human magnetic resonance imaging**. T2-weighted axial magnetic resonance images from patients managed at the St. Louis Children's Hospital Neurofibromatosis Clinical Program (D.H.G.) were de-identified. The MRI scans were de-identified images from children cared for at the St. Louis Children's Hospital NF Clinical Program. No IRB consent is required, as the use of anonymized human samples is not considered human subjects research by the Washington University IRB.

**Mouse magnetic resonance imaging**. Mice were intraperitoneally injected with 70 mM MnCl$_2$ in 100 mM Bicine, pH 7.4 at 140 mg/Kg body weight (10x μL) 16–24 h before imaging. Mice were then subcutaneously injected with 0.5 mL 0.9% saline to minimize magnesium toxicity, and half of the mouse cage was placed on a medium heating pad setting. T1-weighted images (T1W) were collected as previously reported[55]. T1W were collected with 2D spin-echo (SEMS, at 4.7 T): TR/TE, 300 ms/11 ms; matrix size, 128 × 128; FOV, 16 × 16 mm$^2$; slice thickness, 0.5 mm; averages, 16 × 2 (10 × 2 min.). In addition, the 3D T1W were collected using a gradient-echo sequence (GE3D): Matrix size, 128 × 128 × 128; FOV, 24 × 24 × 24 mm$^3$; TR/TE, 4 ms/2 ms; Flip angle, 12°; averages, 8 (9 min). All image data were zero-padded and applied with a Gaussian filter (Sigma 0.75) before conversion to a NIfTI format for analysis.

**Immunohistochemistry, H&E, and PAS staining**. *Nf1*[OPG] mice were euthanized and transcardially perfused with Ringer's solution, followed by 4% paraformaldehyde (PFA). Optic nerves and lungs were harvested and processed for paraffin embedding and sectioning in either the Ophthalmology (optic nerves) or Pulmonary Morphology (lungs) core facilities. H&E and PAS staining were performed as previously reported[8,12,52,53]. Serial 4-μm paraffin sections of the optic nerve were immunostained with the antibodies listed in Supplementary Table 1. Development was performed using the Vectastain ABC kit (Vector Laboratories, PK4000), and imaged for quantitation in Zeiss AxioScan-Z1 and Leica ICC50W microscope using LAS EZ software.

**T cell isolation**. Mouse spleens were homogenized into single-cell suspensions by digesting in PBS containing 0.1% BSA and 0.6% Na-citrate, washed, and incubated with 120 Kunitz units of DNase I for 15 min following red blood cell lysis (eBioscience, 00433357)[12]. Cells were then filtered through a 30-μm cell strainer to generate a single-cell suspension. CD4[+] and CD8[+] T cells were isolated using CD8a (Miltenyi Biotec, 130-104-075) and CD4 (Miltenyi Biotec, 130-117-043) T cell isolation kits, respectively. T cells were maintained at $2.5 \times 10^6$ cells/ml in RPMI-1640 medium supplemented with 10% FBS and 1% P/S. T cells were activated using 1.25 μg/ml anti-mouse CD3 (Fisher Scientific, 16-0031-85) and 2 μg/ml

anti-mouse CD28 (Fisher Scientific, 16-0281-82) antibody treatment for 2d. Recombinant mouse midkine (R&D Systems, 9760-MD-050) was added, and Ccl4 levels were quantitated by ELISA (R&D Systems, MMB00). Conditioned medium was collected from activated T cells for chemokine assays and microglia co-culture experiments. Cell viability was measured using a commercial WST-1 cell viability assay (Millipore).

**Microglia isolation**. Microglia were isolated from mouse brains as previously described[12]. Briefly, mouse brains were perfused with D-PBS (Fisher Scientific, 14287072), and single-cell suspensions were generated using the multi-tissue dissociation kit (Miltenyi-130-110-201), as per the manufacturer's protocol. The resulting cells were maintained in minimal essential medium supplemented with 1 mM L-glutamine, 1 mM sodium pyruvate, 0.6% D-(+)-glucose, 1 ng/ml GM-CSF, 100 μg/ml P/S, and 10% FBS for 2 weeks. Microglia were separated from astrocytes by shaking (200 g, 5 h, 37 °C) after 2 weeks. In total, $5 \times 10^5$ microglia were grown in T cell-conditioned media (TCM) for 48 h[8,9], followed Ccl5 (R&D Systems, MMR00) and decorin (Abcam, ab155454) determinations by ELISA. Mouse recombinant Decorin protein (R&D Systems, 1060-DE-100), AZ084(CCR8IN; MedChemExpress, CAS No-929300-19-6), Maraviroc (CCR5IN; MedChemExpress, CAS no-376348-65-1) was added to the TCM microglia co-culture, and Ccl5 levels were quantified by ELISA. Mouse recombinant CCL4 (R&D Systems, 451-MB-010/CF), Mouse recombinant Biglycan (R&D Systems, 8128-CM-050), Mouse recombinant TGFβ1 (R&D Systems, 7666-MB-005), Caffeic acid phenethyl ester (NFκB-IN; Fisher Scientific, 274310) was added to the Viability were added to the microglia and Ccl5 levels quantified by ELISA. Viability was determined using WST-1 assay reagent (Abcam, ab155902).

**Phagocytosis assay**. Phagocytosis was determined using green fluorescent latex beads (Sigma, L1030-1ML) pre-opsonized in fetal bovine serum (FBS) (1:5 ratio) for 1 h at 37 °C at a ratio of 1:5 before dilution to a final concentration of 0.01% (v/v) and 0.05% (v/v) in DMEM. Purified microglia were plated at a density of 50,000 cells/cm$^2$, as previously reported[11]. After 24 h, the cells were incubated with fluorescent latex beads in FBS for 1 h at 37 °C, washed, and then fixed in 4% PFA for 15 min. Microglia were blocked prior to incubation with primary antibody (Iba1,1:200 Wako Chemicals 019-1794, Japan) overnight at 4 °C, washed, and then incubated with secondary antibody for 1 h. Images were acquired on a Leica DFC3000G fluorescent microscope using a ×10 objective with associated Leica Application Suite X software. The number of phagocytic cells with beads relative to the total number of cells was calculated and expressed as a percentage: $N_{\text{cell with beads}}/N_{\text{total}}$.

**RNA extraction and real-time qRT-PCR**. RNA was isolated from cells and transcardially perfused optic nerves using the NucleoSpin® RNA Plus kit (Takara-740984.205), as per the manufacturer's instructions. Total RNA was then reverse transcribed into cDNA using the Applied Biosystems High-Capacity cDNA Reverse Transcription Kit (#4374967), as per the manufacturer's instructions. Real-time quantitative PCR (qPCR) was performed by TaqMan gene expression (Supplementary Table 2). ΔΔCT values were calculated using *Gapdh* as an internal control.

**RNA sequencing and analysis**. RNA from PBS- and OVA-treated spleen CD3[+] T cells was isolated and sequenced on an Illumina HiSeq platform. Base calls and demultiplexing were performed with Illumina's bcl2fastq software, and a custom python demultiplexing program with a maximum of one mismatch in the indexing read. The analysis was generated using Partek Flow software, v9.0. RNA-seq reads were aligned to the Ensembl release 99 top-level assembly with STAR v2.6.1d. Gene counts and isoform expression were derived from Ensembl output. Sequencing performance was assessed for the total number of aligned reads, the total number of uniquely aligned reads, and features detected[56]. Normalization size factors were calculated for all gene counts by median ratio. Differential genetic analysis was then performed using DESeq2 to analyze for differences between conditions[57]. Results

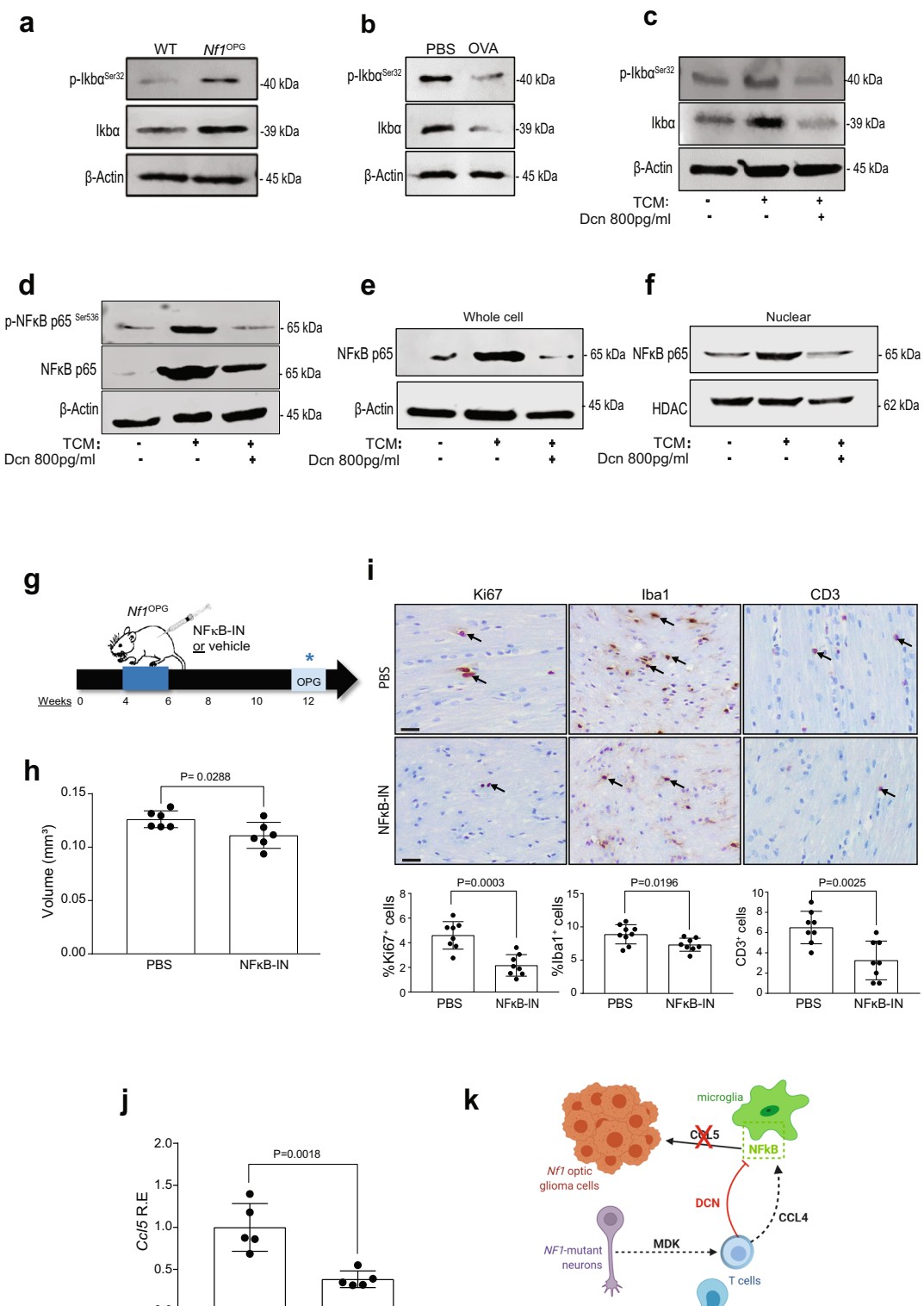

for OVA samples compared separately with controls were filtered for only those genes with P values ≤ 0.01, false discovery rates ≤ 0.05, and log fold changes ≥ 5.

**Human microarray analysis.** Raw Affymetrix U133 Plus 2.0 microarray data were obtained from the Gene Expression Omnibus (GSE31773) derived from samples of circulating CD4⁺ and CD8⁺ T cells from non-severe asthma patients (perfect control of their asthma with 0–1000 µg/d inhaled beclomethasone or an equivalent dosage; n = 4; donors 1–4) and healthy control (n = 5; donors 1, 3, 4, 6, 8)

subjects[58]. The analysis was performed using Partek Flow software, v9.0. mRNA was aligned to the Ensembl release 100 top-level assembly with STAR v2.7.3a. Gene counts and isoform expression were derived from Ensembl output. Non-coding RNA data were excluded from further analysis. Normalization size factors were calculated for all gene counts by CPM to adjust for differences in library size. Gene-specific analysis was performed using the lognormal with shrinkage model (limma-trend method) to analyze for differences between disease and cell-type conditions. The GSA (gene-specific analysis) results were filtered for the DCN gene. The cut-off used for significance was P value ≤ 0.05.

**Fig. 6 Decorin suppresses _Nf1_-OPG formation through microglia NFkB inhibition. a** _Nf1_[OPG] mouse optic nerves ($n = 3$) have increased IκbΑ phosphorylation relative to WT mice ($n = 3$). **b** The increased IκbΑ phosphorylation in PBS-treated _Nf1_[OPG] mouse optic nerves ($n = 3$) was reduced by OVA treatment ($n = 3$). **c** Activated TCM (act-Tm) increased IκbΑ phosphorylation in microglia, which was reduced following the addition of decorin (800 pg/ml). **d** Decorin (800 pg/ml) blocks the increased p65-NFκB Ser[536] phosphorylation induced by activated TCM (act-Tm) treatment ($n = 3$). **e** Decorin (800 pg/ml) blocks the increased total p65-NFκB expression, (**f**) as well as the nuclear localization of p65-NFκB, induced by activated TCM (act-Tm) treatment ($n = 3$). β-actin and HDAC are used as controls for total protein expression and the nuclear fractions, respectively. **g** Schematic representation of the NFκB inhibitor treatment used. _Nf1_[OPG] mice were treated between 4 and 6 weeks of age with the CAPE NFκB inhibitor ($n = 8$), whereas control _Nf1_[OPG] mice received PBS only ($n = 8$). Isolated optic nerves were analyzed at 12 weeks of age. **h** NFκB inhibitor treatment reduced optic glioma volume and (**i**) proliferation (%Ki67[+] cells), as well as microglia (%Iba1[+] cells) and T-cell (CD3[+]) content within the optic nerves of _Nf1_[OPG] mice relative to vehicle-treated controls. Two-tailed Student's _t_ test. **j** Reduced _Ccl5_ RNA expression was observed in the optic nerves from _Nf1_[OPG] mice treated with the CAPE NFκB inhibitor (NFκB-IN) ($n = 5$). Two-tailed Student's _t_ test. **k** Proposed model of asthma-induced decorin suppression of the _Nf1_[OPG] neuron-immune-cancer cell axis. Asthma induces T cell production of decorin, which reduces T cell Ccl4-mediated microglia Ccl5 expression through inhibition of NFκB signaling. Data are presented as the means ± SEM. Exact _P_ values are indicated within each panel. **i** Scale bars 40 μm. From left to right in each panel: **h** $P = 0.0288$; **i** $P = 0.0003$, $P = 0.0196$, $P = 0.0025$; **j** $P = 0.0018$.

**Western blotting**. Total protein was extracted using RIPA buffer and quantified using the Pierce BCA protein assay kit (Fisher scientific, PI23225). Subcellular fractions were isolated following the manufacturer's protocol (Thermo scientific 78840). In all, 40 μg of total protein lysate was separated in precast SDS-polyacrylamide gels by electrophoresis and transferred onto PVDF membranes, followed by blocking and incubation with the indicated antibody (Supplementary Table 1) overnight. Proteins were detected with IRDye-conjugated secondary antibodies in LI-COR Odyssey Imaging system using Image Studio v5.2.

**Statistical analysis**. Data were analyzed using GraphPad Prism software. To determine differences between two groups, two-tailed Student's _t_ test was used whereas multiple comparisons were analyzed by one-way analysis of variance (ANOVA) with Dunnett's multiple comparisons test. Statistical significance was set at $P \le 0.05$. All experiments were independently repeated at least three times.

**Reporting summary**. Further information on research design is available in the Nature Research Reporting Summary linked to this article.

## Data availability

The RNA-seq data were deposited in GEO (Accession number 166551). The Raw Affymetrix U133 Plus 2.0 microarray data used in this study are available in the GEO database under accession code GSE31773. The remaining data are available within the Article, Supplementary Information or Source Data file. Source data are provided with this paper.

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

## Acknowledgements

This work is funded by a grant from the National Institute of Neurological Disorders and Stroke (1-R35-NS07211-01 to D.H.G.) and National Heart, Lung, and Blood Institute (R35-HL145242 to M.J.H.). E.C. was supported by NIH T35 NHLBI Training Grant (5-T35-HL007815) and a POST grant from Alex's Lemonade Stand Foundation. We thank the Genome Technology Access Center in the Department of Genetics at Washington University School of Medicine for help with genomic analysis. The authors thank the Ophthalmology and Pulmonary Morphology Cores at the Washington University School of Medicine for tissue processing. The Washington University Ophthalmology Core facility support is supported by funding from the National Eye Institute (P30EY002687), while the Washington University Genome Engineering and iPSC Core Center is subsidized by funding from an NCI Cancer Center Support Grant (P30-CA091842) and by ICTS/CTSA Grant# UL1TR002345 from the National Center for Research Resources (NCRR), a component of the National Institutes of Health (NIH), and NIH Roadmap for Medical Research. This publication is solely the responsibility of the authors and does not necessarily represent the official view of NCRR or NIH. We thank Washington University Center for Cellular Imaging (WUCCI) supported by Washington University School of Medicine, The Children's Discovery Institute of Washington University and St. Louis Children's Hospital (CDI-CORE-2015-505 and CDI-CORE-2019-813), and the Foundation for Barnes-Jewish Hospital (3770 and 4642). Figure illustrations were created with BioRender.com.

## Author contributions

Conceptualization: J.C. and D.H.G.; methodology: J.C. and M.J.H.; writing: J.C. and D.H.G.; investigation: J.C. performed most of the experiments; S.S., A.K.G., and E.C. performed some of the mouse and immunohistochemistry experiments; O.C. performed the mouse RNA sequencing analysis; A.B. performed the human RNA sequencing analysis; X.G. and J.R.G. performed the mouse MRI analysis. Resources: D.H.G.; supervision: D.H.G.; funding: D.H.G.

## Competing interests

D.H.G. has a licensing agreement with the Tuberous Sclerosis Alliance (GFAP-Cre mice). J.C., S.S., A.K.G., E.C., X.G., J.R.G., and M.J.H. declare no competing interests.
