## [Peer Review File · Nature Communications]

Reviewers' Comments:

Reviewer #2:

Remarks to the Author:

This is a highly interesting manuscript exploring the clinically relevant inverse association of asthma and gliomas using a preclinical model of optic nerve glioma. The study is based on epidemiological observations and previous studies supporting a key role of T cell-derived CCL4 in regulating microglial CCL5 thus supporting the growth of optic nerve glioma. In this study the authors identify decorin to be a key suppressor of microglial CCL5 and suppressor of optic nerve glioma growth. These results identify a novel mediator of optic nerve glioma growth and thus not only provide a mechanistic link between asthma and glioma growth but also provide a novel therapeutic target. The paper is well written and thoroughly structured. I would suggest to address the following points in order to strengthen the mechanistic link between asthma and glioma growth.

1. Both, in the model and in NF1 patients there heterozygosity for a germline Nf1 gene mutation. NF1 mutations may impact immune function independent of gliomagenesis. Is allergic asthma in the preclinical model and in patients more prevalent and/or more severe than in patients / mice without heterogeneous NF1 mutations? This is particularly important as decorin is implicated in regulating severity of asthma.
2. The data implies decorin to be upregulated in allergic asthma in T cells, thus blocking CCL5 production from microglial cells in an NFkB-dependent fashion. Is decorin also regulated by midkine and thus may function as a counter-mechanism contributing to optic nerve glioma growth also in the absence of asthma?
3. Decorin has been shown to promote asthma by suppressing Tregs. Is asthma or treatment with decorin associated with a reduced number of Tregs?
4. A key function of decorin is to sequester and reduce TGF-beta bioavailability. Is TGF-beta involved in the regulation of CCL5 in microglia?

Reviewer #3:

Remarks to the Author:

In this study, Chatterjee and colleagues describe the effects and mechanisms of the ovalbumin and house dust mice-induced allergic reaction in the formation of optic nerve gliomas in mice. It is potentially an interesting study. However, there are several major issues that need to be addressed.

Major comments:

- 1) The study is based on the group's 2016 report on a non-significant inverse association between asthma and optic gliomas in pediatric patients. It is not clear if there is any more information on whether NF1 patients with asthma have delayed development/diagnosis of optic glioma, or if the NF1 optic glioma patients have dysfunctional T cells and/or CCL4 levels in the blood samples?
- 2) Appropriate methods need to be used to measure tumor size. In addition, data should be provided to confirm tumor formation and to evaluate tumor growth after treatment. Specifically, in Fig 1, the authors mention that by 12-wk of age, optic gliomas are evident in >95% of mice, and OVA/HDM treatment demonstrated 'durable effects on gliomagenesis'. However, only optic nerve volume data was provided. MRI imaging, or histological evidence of glioma formation should be provided. In Fig 4, the authors treated mice with decorin and NFkB inhibitor, and claimed that reduced Ki67+ cells are 'suppressed' (Fig 4b) and they 'blocked' (Fig 4f) optic glioma formation in vivo. The optic nerve volume, MRI imaging, and/or histological data should be provided to demonstrate changes in tumor growth.
- 3) The study focused on microglia expression of CCL5 in response to Ova/HDM, decorin and NFkB inhibitor treatments. There are several flaws in this study design. First, the authors used Iba1 staining for evaluating the number of microglia. Iba1 is expressed in both microglia and macrophage. Markers distinguishing the 2 cell types (such as CCR2 and CX3CR1) should be used

to determine whether the changes in the number of Iba1+ cells is a result of changes in microglia expansion or from changes in the recruitment of macrophages. Second, changes in the number of Iba1+ cells might not translate to functional changes. How the OVA/HDM, decorin and NFkB inhibitor treatments affect the function of microglia and macrophages should be addressed. Does OVA/HDM asthma-induction treatment induce a switch between the pro-inflammatory M1 and pro-tumor M2 phenotype (in Fig 1-3)? NFkB affects multiple aspects of inflammation and tumor progression. How does the NFkB inhibitor affect the microglia and macrophages phenotype and function (in Fig 4)?

4) The authors concluded that 'decorin inhibits microglia CCL5 production by NFkB inhibition'. There are several problems with this study design and conclusion. First, the authors concluded that NFkB inhibitor reduced microglia Ccl5 production, but only provided Ccl5 RNA data in the optic nerve chunk tissue. A more detailed dissection of the NFkB effects on Ccl5 expression in the source cell is needed. Second, the authors showed that decorin reduced T cell conditioned medium-induced Ikb phosphorylation in the microglia cells in vitro, and NFkB inhibitor reduced the number of Ki67+ cells. From these 2 parallel studies, the conclusion that decorin functions via NFkB inhibition can not be drawn.

Comments on data:

Figure 1

- 1) In the control (PBS treated) animals, the authors should explain why the optic nerve volume decreased significantly from week 12 to week 24 (Fig 1b and Fig 1e).
- 2) Fig 1f, the authors showed durable effects 24 weeks after OVA treatment. What is the extended effect from HDM, at 24 weeks?
- 3) Extended data Fig 1d, what is the time point the MRI was performed? How does the NF1OPG optical nerve volume compare in WT versus the NF1+/- mice?
- 4) Extended data Fig 1e, Y-axis label of volume data should be mm³.
- 5) Extended data Fig 1f, middle panel shows thickened optical nerves. However, this middle panel image is significantly larger than the left and right panels, scale bars should be added.
- 6) Extended data Fig 1h, showing 9 dots in the Nf1+/- group, while the legend states n=8.
- 7) From the authors' previous publication Guo et al. 2020 I infer that the WT mice used in this manuscript are Nf1flox/flox mice. Please mention this explicitly in the manuscript. Also, please state/speculate re. the potential reasons for the increase in tumor volume with PBS in NF1OPG mice compared to the tumor volume of the WT mice.
- 8) Figure legends stating the conclusions is unusual and not the format of Nature journals. The authors' previous paper in Nature Communications by Guo et al. 2020 does not have this format. Figure legends may need to be reformatted to follow the Nature Comm guidelines.

Figure 2

- 1) Fig 2b. The authors concluded that 'stimulated T cells produced the highest level of Ccl4' (p4, line 92). What is the timing of tissue collection for the qPCR study? Is the tissue collected after glioma formation? If a tumor is formed, then the high Ccl4 RNA level detected in the optic nerve, as compared to those in the lymph node and blood, could come from multiple cells. For example, tumor cells, as well as macrophage and Schwann cells are known to produce Ccl4.
- 2) Fig 2c was not referenced in the text.
- 3) Extended Fig 2b-c, the result section stated 'ccl4 and ccl5 were examined at 12 and 24 weeks of age' and referenced Extended Fig 2b-c (p4, line 96), yet that figure showed data only at one time point, and did not specify which time point it is.
- 4) For the same experiment with OVA the authors use n=4 mice per group at 12 weeks (Fig 2e), but only use 3 mice per group at 24 weeks (Extended data Fig 2e). Are mice dying between 12 and 24 weeks in both the groups?

Figure 3

- 1) Fig 3a. 'inhibitor Decorin' was not referenced in Fig 3a, while it was described in the figure legend.

Figure 4

- 1) Fig 4b, did decorin treatment change the optical nerve volume?
- 2) Fig 4c-e: Please provide the NFkB level/activity, e.g., p65 phosphorylation and nuclei translocation and DNA binding.
- 3) Fig 4f, did NF-kB inhibitor change optical nerve volume?
- 4) Fig 4f, NFkB directly reduced the number of microglia, does it inhibit microglia expansion or reduce microglia viability?

Reviewer #4:

Remarks to the Author:

The article by Chatterjee et al describes a novel mechanism underlying the clinical observation that children with asthma have a reduced incidence of brain tumour, in particular optic gliomas. They identify a putative role for decorin, a small leucine rich proteoglycan, which deposition has been shown to be altered in asthma, and which deficiency in a mouse model leads to a mitigated response to allergen sensitization and challenge. The authors define a mechanistic pathway for the effect of decorin, implicating Cc15 production and NFkB inhibition, and leading to reduced glioma formation in a well characterized pre-clinical mouse model of low grade glioma. This is a very interesting study, crossing disciplines and opening up a new line of inquiry. The experimental approach is novel, well developed and has potential significance for an important clinical problem.

Major comments

Animal models are often problematic. The authors describe "asthma induction" in mice, but the ovalbumin protocol they use differs from those cited (ref 17, 18 and 50). One ovalbumin model cited included sensitization with adjuvant x 2, prior to repeated aerosol administration (x9) to induce an "asthmatic" phenotype; the other model describes both acute and chronic challenges, again with more repeated challenges. In the current manuscript, mice were sensitized x 2 and then challenged x 4. Given the differences in the model, the authors should have more carefully described the "asthma-like" changes induced. They show representative images of bronchial walls, with evidence of thickening. This is not sufficient. There are no physiologic measurements, no demonstration of enhanced responsiveness, no characterization of the bronchial wall thickening (whether airway smooth muscle, reticular basement membrane), no morphometric measurements, no information about inflammatory cell infiltration (eg. eosinophils), etc. Given the induction of an "asthma-like phenotype" is key, establishing the phenotype is crucial to the subsequent experimentation. For example, the authors' claim "asthma blocks optic glioma formation" (lines 83-84).

The RNA sequencing results (Fig 3j) are very interesting and point to decorin as the molecule of interest. Biglycan is another SLRP with similar/overlapping actions as decorin. It would be interesting to know if biglycan was altered in this model system.

The published data on changes in decorin in human asthma are inconsistent. Huang et al (AJRCCM, 160:725,1999) showed no difference in decorin deposition in the airway walls of mild atopic asthmatics, as did Pini et al (AJP Lung: Cell Mol Physiol, 290:1052,2006) in the airway walls of moderate and severe asthmatics. On the other hand, de Medeiros Matsushita et al (J Pathol, 207:102, 2005) showed a decrease in decorin in the airway wall of patients dying with severe asthma. As such, the authors data (fig 3m) showing an increase in Dcn expression in CD4 and CD8 cells is not in keeping with that previously reported in the literature. This warrants some discussion. Can the authors provide more information about the patient population? (ref 55 reports on an animal model). Can the authors reconcile the lymphocyte data showing an increase in Dcn with the airway wall data showing stable or decreased decorin deposition?

Minor comments

Did the authors look at cytokines that are typically altered in asthma (IL4, IL5, IL3) to determine

their potential impact on the chemokines of interest?

Mice were administered decorin and the NF κ B inhibitor by ip injection every other day for 2 weeks between 4 and 6 weeks of age. How did the authors choose this mode/dose of treatment? Is 1 mg/ml within the physiologic range? Were there off target effects? Why was the Dcn effect more specific than the NF κ B effect (fig 4f)?

The statistical approach is appropriate.

The text is clear and accessible.

Line 192: References 28 and 29 do not report data in asthma.

Ref 30 is the same as ref 17.

Reviewer #2

This is a highly interesting manuscript exploring the clinically relevant inverse association of asthma and gliomas using a preclinical model of optic nerve glioma. The study is based on epidemiological observations and previous studies supporting a key role of T cell-derived CCL4 in regulating microglial CCI5 thus supporting the growth of optic nerve glioma. In this study the authors identify decorin to be a key suppressor of microglial CCI5 and suppressor of optic nerve glioma growth. These results identify a novel mediator of optic nerve glioma growth and thus not only provide a mechanistic link between asthma and glioma growth but also provide a novel therapeutic target. The paper is well written and thoroughly structured. I would suggest to address the following points in order to strengthen the mechanistic link between asthma and glioma growth.

1. Both, in the model and in NF1 patients there heterozygosity for a germline Nf1 gene mutation. NF1

mutations may impact immune function independent of gliomagenesis. Is allergic asthma in the preclinical model and in patients more prevalent and/or more severe than in patients / mice without heterogeneous NF1 mutations? This is particularly important as decorin is implicated in regulating severity of asthma. We appreciate the reviewer's comment, which we address in two ways. First, there are few human studies examining asthma in individuals with NF1. In one study¹, the prevalence of asthma was higher in Japanese adults with NF1 relative to a control group of subjects, whereas a questionnaire-based study found a higher incidence of asthma (27% versus 10%)². Similarly, higher serum IgG levels have been reported in individuals with NF1³. Since these studies focused on adults, beyond the age at which optic gliomas occur in children (mean 4.5 years), we took a second approach by comparing the severity of asthma induced by HDM and OVA in wild type and *Nf1*^{+/-} mice. These new data (revised Extended Data Fig. 1a, b) show similar immune infiltration and alveolar pathology in the lungs and similar serum IgE levels (revised Extended Data Fig. 1c).

Extended Data Fig. 1. WT and *Nf1*^{+/-} mice were treated between 4-6 weeks of age with either OVA or HDM

(experimental asthma) or PBS (controls). Representative (a) H&E and (b) PAS-stained images of the lungs from WT and *Nf1*^{+/-} mice treated with PBS (*n*=6), HDM (*n*=6) and OVA (*n*=6). Increased alveolar wall thickening and immune cell infiltration were observed 7 days after OVA and HDM treatment. Boxes indicate thickened bronchial walls. Small scale bar, 500 μm; large scale bar, 250 μm. (c) Similar increases in serum IgE were observed in WT (*n*=4) and *Nf1*^{+/-} (*n*=4) mice treated with OVA and HDM relative to PBS-treated control mice. Data are presented as the mean ± SEM. One-way ANOVA with Bonferroni post-test correction; n.s., not significant; Exact P values are indicated within each panel. From left to right in each panel: (c) ns, ns, ns, P<0.0001.

2. The data implies decorin to be upregulated in allergic asthma in T cells, thus blocking CCL5 production from microglial cells in an NFκB-dependent fashion. Is decorin also regulated by midkine and thus may function as a counter-mechanism contributing to optic nerve glioma growth also in the absence of asthma? To address this question, we examined the ability of midkine to induce microglia Ccl5 levels. No induction of Ccl5 was observed following midkine exposure (revised Extended Data Fig. 4b).

Extended Data Fig. 4b. Midkine (100ng/ml) treatment of WT and *Nf1*^{+/-} microglia (*n*=4) for 48 h does not increase Ccl5 production.

3. Decorin has been shown to promote asthma by suppressing Tregs. Is asthma or treatment with decorin associated with a reduced number of Tregs? To address this important issue, we examined the optic gliomas from *Nf1*-OPG mice treated with HDM and OVA, and did not observe any Foxp3⁺ cells (Tregs) in these tumors (revised Extended Data Fig. 4d).

Extended Data Fig. 4d. No Foxp3⁺ cells were detected in the optic nerves from 12-week-old *Nf1*^{OPG} mice treated with PBS, OVA or HDM.

4. A key function of decorin is to sequester and reduce TGF-beta bioavailability. Is TGF-beta involved in the regulation of CCL5 in microglia? To determine whether TGF-beta could induce Ccl5 in microglia, we treated microglia with TGF-beta, but did not observe any increase in Ccl5 expression (Extended Data Fig. 4e).

Extended Data Fig. 4e. WT microglia were treated with either TGFβ (5ng/ml) alone or in combination with Ccl4 (6000pg/ml). TGFβ does not induce microglia Ccl5 production, either alone or in combination with

Reviewer #3

In this study, Chatterjee and colleagues describe the effects and mechanisms of the ovalbumin and house dust mice-induced allergic reaction in the formation of optic nerve gliomas in mice. It is potentially an interesting study. However, there are several major issues that need to be addressed.

1) The study is based on the group’s 2016 report on a non-significant inverse association between asthma and optic gliomas in pediatric patients. It is not clear if there is any more information on whether NF1 patients with asthma have delayed development/diagnosis of optic glioma, or if the NF1 optic glioma patients have dysfunctional T cells and/or CCL4 levels in the blood samples? Unfortunately, there are no data to demonstrate that NF1 patients with optic gliomas have dysfunctional T cells or altered CCL4 levels in the blood. Similarly, there are few human studies examining asthma in individuals with NF1. In one study¹, the prevalence of asthma was higher in Japanese adults with NF1 relative to a control group of subjects, whereas a questionnaire-based study found a higher incidence of asthma (27% versus 10%)². Similarly, higher serum IgG levels have been reported in individuals with NF1³. Since these studies focused on adults, beyond the age at which optic gliomas occur in children (mean 4.5 years), we took a second approach by comparing the severity of asthma induced by HDM and OVA in wild type and *Nf1*^{+/-} mice. These new data (revised Extended Data Fig. 1a-c) show similar immune infiltration and alveolar pathology in the lungs and similar serum IgE levels.

Extended Data Fig. 1. WT and *Nf1*^{+/-} mice were treated between 4-6 weeks of age with either OVA or HDM (experimental asthma) or PBS (controls). Representative (a) H&E and (b) PAS-stained images of the lungs from WT and *Nf1*^{+/-} mice treated with PBS (n=6), HDM (n=6) and OVA (n=6). Increased alveolar wall thickening and immune cell infiltration were observed 7 days after OVA and HDM treatment. Boxes indicate thickened bronchial walls. Small scale bar, 500 mm; large scale bar, 250 mm. (c) Similar increases in serum IgE were observed in WT (n=4) and *Nf1*^{+/-} (n=4) mice treated with OVA and HDM relative to PBS-treated control mice. Data are presented as the mean ± SEM. One-way ANOVA with Bonferroni post-test correction; n.s., not

significant; Exact P values are indicated within each panel. From left to right in each panel: (c) ns, ns, ns, P<0.0001.

2) Appropriate methods need to be used to measure tumor size. In addition, data should be provided to confirm tumor formation and to evaluate tumor growth after treatment. Specifically, in Fig 1, the authors mention that by 12-wk of age, optic gliomas are evident in >95% of mice, and OVA/HDM treatment demonstrated ‘durable effects on gliomagenesis’. However, only optic nerve volume data was provided. MRI imaging, or histological evidence of glioma formation should be provided. In Fig 4, the authors treated mice with decorin and NFkB inhibitor, and claimed that reduced Ki67+ cells are ‘suppressed’ (Fig 4b) and they ‘blocked’ (Fig 4f) optic glioma formation in vivo. The optic nerve volume, MRI imaging, and/or histological data should be provided to demonstrate changes in tumor growth. We appreciate the reviewer’s point. Similar to their human counterparts, mouse *Nf1*-optic gliomas are scored by the presence of tissue architectural distortion (mass effect, increase in volume), microglia infiltration (*Iba1*⁺ cells), and increased proliferation (Ki67⁺ cells). The human tumors are always compared to normal tissues, rather than to optic nerves from patients with *NF1*. In this regard, we and others have described tortuosity of the optic nerves in children without optic gliomas, similar to what we report herein for *Nf1*^{+/-} mice by direct measurements and magnetic resonance imaging (revised Extended Data Fig. 2c). Importantly, *Nf1*^{+/-} mice have optic nerves with increased microglia content and volumes that cannot be distinguished from *Nf1*-OPG mice.

Extended Data Fig. 2c. Thickened optic nerves were observed in *Nf1*^{+/-} mice by MnCl₂-enhanced T1-weighted magnetic resonance imaging and area measurements. Red arrows point to the optic nerves.

For this reason, proliferation is the defining difference between non-neoplastic optic nerves from *Nf1*^{+/-} mice and optic gliomas. As suggested by the reviewer, we now provide all data (T cell/microglia content, %Iba1⁺ cells, and volumes) in the revised Fig. 6.

Fig. 6. (a) *Nf1*^{OPG} mouse optic nerves (*n*=3) have increased IκBα phosphorylation relative to WT mice (*n*=3). (b) The increased IκBα phosphorylation in PBS-treated *Nf1*^{OPG}

mouse optic nerves ($n=3$) was reduced by OVA treatment ($n=3$). (c) Activated TCM (act-Tm) increased I κ b α phosphorylation in microglia, which was reduced following the addition of decorin (800pg/ml). (d) Decorin (800 pg/ml) blocks the increased p65-NF κ B Ser⁵³⁶ phosphorylation induced by Ccl4 treatment ($n=3$). (e) Decorin (800 pg/ml) blocks the increased total p65-NF κ B expression, (f) as well as the nuclear localization of p65-NF κ B, induced by Ccl4 treatment ($n=3$). β -actin and HDAC are used as controls for total protein expression and the nuclear fractions, respectively. (g) Schematic representation of the NF κ B inhibitor treatment used. *Nf1*^{OPG} mice were treated between 4 and 6 weeks of age with the CAPE NF κ B inhibitor ($n=8$), whereas control *Nf1*^{OPG} mice received PBS only ($n=8$). Isolated optic nerves were analyzed at 12 weeks of age. NF κ B inhibitor treatment reduced (g) optic glioma volume and (h) proliferation (%Ki67⁺ cells), as well as microglia (%Iba1⁺ cells) and T cell (CD3⁺) content within the optic nerves of *Nf1*^{OPG} mice relative to vehicle-treated controls. Two-tailed Student's *t*-test. (i) Reduced *Ccl5* RNA expression was observed in the optic nerves from *Nf1*^{OPG} mice treated with the CAPE NF κ B inhibitor (NF κ B-IN) ($n=5$). (j) Proposed model of asthma-induced decorin suppression of the *Nf1*^{OPG} neuro-immune-cancer cell axis. Asthma induces T cell production of decorin, which reduces T cell Ccl4-mediated microglia *Ccl5* expression through inhibition of NF κ B signaling. All data are presented as the mean \pm SEM. Exact p-values are indicated within each panel. (h) Scale bars 40 μ m. From left to right in each panel: (g) P=0.0288; (h) P=0.0003, P=0.0196, P=0.0025; (i) P=0.0018.

3) The study focused on microglia expression of CCL5 in response to Ova/HDM, decorin and NF κ B inhibitor treatments. There are several flaws in this study design. First, the authors used Iba1 staining for evaluating the number of microglia. Iba1 is expressed in both microglia and macrophage. Markers distinguishing the 2 cell types (such as CCR2 and CX3CR1) should be used to determine whether the changes in the number of Iba1+ cells is a result of changes in microglia expansion or from changes in the recruitment of macrophages. Second, changes in the number of Iba1+ cells might not translate to functional changes. How the OVA/HDM, decorin and NF κ B inhibitor treatments affect the function of microglia and macrophages should be addressed. Does OVA/HDM asthma-induction treatment induce a switch between the pro-inflammatory M1 and pro-tumor M2 phenotype (in Fig 1-3)? NF κ B affects multiple aspects of inflammation and tumor progression. How does the NF κ B inhibitor affect the microglia and macrophages phenotype and function (in Fig 4)? To address these comments, we performed several experiments. First, we used a microglia-specific marker (*Tmem119*)⁴ to demonstrate that the Iba1⁺ cells in these tumors are *Tmem119*⁺ microglia, rather than macrophages (revised Extended Data Fig. 4k). Second, we examined the expression of conventional M1 and M2 markers following OVA/HDM induction, and saw no changes (revised Extended Data Fig. 3h-l). Third, asthma induction does not alter microglia phagocytosis or proliferation (revised Extended Data Fig. 4m, n). Fourth, spleen macrophages do not induce *Ccl5* in response to Ccl4 (revised Extended Data Fig. 4l).

Extended Data Fig. 4. (k) 12-week-old *Nf1*^{OPG} optic nerves ($n=5$) following PBS, OVA, and HDM treatment were co-labelled with Iba1 and *Tmem119* antibodies. Nearly all of the Iba1⁺ cells were *Tmem119*⁺ microglia, rather than

Tmem119-negative macrophages. **(I)** Mouse splenic macrophages failed to induce Ccl5 expression in response to Ccl4. OVA treatment of mice does not change microglia **(m)** phagocytosis or **(n)** viability ($n=4$, $66.88 \pm 4.09\%$ phagocytic cells). Scale bar, 25 μ m. A one-way ANOVA with Bonferroni post-test correction was used.

Extended Data Fig. 3. OVA- and PBS-treated *Nfl*^{OPG} mouse optic nerves ($n=4$) have similar expression of M1 markers [(h) *Il1 β* , (i) *Il6*, (j) *Tnfa*] and M2 markers [(k) *Il4*, (l) *Il10*]. All data are presented as the mean \pm SEM. A two-tailed Student's *t*-test was used.

4) The authors concluded that 'decorin inhibits microglia CCL5 production by NF κ B inhibition'. There are several problems with this study design and conclusion. First, the authors concluded that NF κ B inhibitor reduced microglia Ccl5 production, but only provided Ccl5 RNA data in the optic nerve chunk tissue. A more detailed dissection of the NF κ B effects on Ccl5 expression in the source cell is needed. Second, the authors showed that decorin reduced T cell conditioned medium-induced I κ b phosphorylation in the microglia cells *in vitro*, and NF κ B inhibitor reduced the number of Ki67+ cells. From these 2 parallel studies, the conclusion that decorin functions via NF κ B inhibition can not be drawn. We appreciate this suggestion to strength our conclusions by including *in vitro* studies demonstrating that decorin reduces Ccl4-mediated microglia Ccl5 expression, and that Ccl4-mediated microglia Ccl5 expression is blocked by NF κ B inhibition (revised Extended Data Fig. 6c, d). In addition, we include further experiments examining NF κ B phosphorylation (revised Fig. 6d) and nuclear localization (revised Fig. 6e-f).

Extended Data Fig. 6. (c) Decorin inhibits increased microglia Ccl5 production in response to Ccl4 treatment *in vitro*. (d) NF κ B inhibition (CAPE) blocks microglia Ccl5 production in response to

Ccl4 treatment *in vitro*.

Fig. 6. (d) Decorin (800 pg/ml) blocks the increased p65-NFκB Ser⁵³⁶ phosphorylation induced by Ccl4 treatment ($n=3$). **(e)** Decorin (800 pg/ml) blocks the increased total p65-NFκB expression, **(f)** as well as the nuclear localization of p65-NFκB, induced by Ccl4 treatment ($n=3$). β-actin and HDAC are used as controls for total protein expression and the nuclear fractions, respectively.

Comments on data:

Figure 1

5) In the control (PBS treated) animals, the authors should explain why the optic nerve volume decreased significantly from week 12 to week 24 (Fig 1b and Fig 1e). We thank the reviewer for pointing out this discrepancy, which is now corrected in the **revised Fig. 1b, e**. There is no difference between the volumes ($P=0.0785$).

Fig. 1b. (left) OVA and HDM treatments reduced optic nerve volumes (PBS, $n=6$; OVA, $n=7$; HDM, $n=8$) of *Nfl*^{OPG} mice relative to the vehicle-treated groups (PBS, $n=8$; OVA, $n=8$; HDM, $n=9$). Reduced optic nerve volumes **(e)** were also observed at 24 weeks of age after OVA treatment **(right)** between 4 and 6 weeks of age (PBS, $n=8$; OVA, $n=8$).

6) Fig 1f, the authors showed durable effects 24 weeks after OVA treatment. What is the extended effect from HDM, at 24 weeks? As suggested, we now include these new data in the **(revised Fig.1f)**.

Fig. 1f. Reduced proliferation (%Ki67⁺ cells; **f**) were also observed at 24 weeks of age after OVA and HDM treatment between 4 and 6 weeks of age (PBS, $n=8$; OVA, $n=8$). Bar graphs represent the means ± SEM.

7) Extended data Fig 1d, what is the time point the MRI was performed? How does the NF1OPG optical nerve volume compare in WT versus the NF1+/- mice? These analyses were performed at 3 months of age (now indicated in the revised Extended Data Fig. 2b,c.

Extended Data Fig. 2b. Optic nerve volumes from *Nf1*^{+/-} (n=8) and *Nf1*^{OPG} mice (n=13) compared to WT controls (n=6).

Extended Data Fig. 2c. Thickened optic nerves were observed in *Nf1*^{+/-} and *Nf1*-OPG mice by MnCl₂-enhanced T1-weighted magnetic resonance imaging and volume measurements. *Nf1*^{+/-} and *Nf1*-OPG mouse optic nerve areas were indistinguishable by MRI. Arrows point to the optic nerves.

8) Extended data Fig 1e, Y-axis label of volume data should be mm³. We appreciate the reviewer pointing this out, the graph represent area and thus represented as mm².

9) Extended data Fig 1f, middle panel shows thickened optical nerves. However, this middle panel image is significantly larger than the left and right panels, scale bars should be added. We appreciate the reviewer pointing this out, which we now correct in the revised Extended Data Fig 2d.

10) Extended data Fig 1h, showing 9 dots in the Nf1+/- group, while the legend states n=8. We appreciate the reviewer pointing this out, which we now correct in the revised Fig 1f.

11) From the authors' previous publication Guo et al. 2020 I infer that the WT mice used this manuscript are Nf1flox/flox mice. Please mention this explicitly in the manuscript. Also, please state/speculate the potential reasons for the increase in tumor volume with PBS in NF1OPG mice compared to the tumor volume of the WT mice. We appreciate the reviewer pointing this out, which we now provide in the revised text (page 13). In addition, we have replaced the tumor volume measurements with more representative measurements in the (revised Fig. 1b).

Fig. 1b. OVA and HDM treatments reduced optic nerve volumes (PBS, n=6; OVA, n=7; HDM, n=8) of *Nf1*^{OPG} mice

relative to the vehicle-treated groups (PBS, n=8; OVA, n=8; HDM, n=9).

12) Figure legends stating the conclusions is unusual and not the format of Nature journals. The authors' previous paper in Nature Communications by Guo et al. 2020 does not have this format. Figure legends may need to be reformatted to follow the Nature Comm guidelines. We appreciate the reviewer pointing this out, which we now correct in the revised Figure Legends.

Figure 2

13) Fig 2b. The authors concluded that 'stimulated T cells produced the highest level of Ccl4' (p4, line 92). What is the timing of tissue collection for the qPCR study? Is the tissue collected after glioma formation? If a tumor is formed, then the high Ccl4 RNA level detected in the optic nerve, as compared to those in the lymph node and blood, could come from multiple cells. For example, tumor cells, as well as macrophage and Schwann cells are known to produce Ccl4. The tissues were from the same mice (blood, lymph nodes, optic nerves) at 3 months of age, which is now included in the revised Fig. 2 legend.

14) Fig 2c was not referenced in the text. We apologize for this inadvertent omission, which is now corrected in the revised manuscript.

15) Extended Fig 2b-c, the result section stated 'ccl4 and ccl5 were examined at 12 and 24 weeks of age' and referenced Extended Fig 2b-c (p4, line 96), yet that figure showed data only at one time point, and did not specify which time point it is. We now provide the times when the optic nerves were collected for Ccl4 and Ccl5 measurements in the revised Extended Data Fig 3.

16) For the same experiment with OVA the authors use n=4 mice per group at 12 weeks (Fig 2e), but only use 3 mice per group at 24 weeks (Extended data Fig 2e). Are mice dying between 12 and 24 weeks in both the groups? Similar to patients with NF1-OPGs, mice do not die from their optic gliomas. Unpublished data from our laboratory reveals no death of mice from optic gliomas at 12-15 months of age. *Nfl*^{+/-} mice after 12-15 months of age are at increased risk of developing pheochromocytoma and leukemia^{5,6,7}.

Figure 3

17) Fig 3a. 'inhibitor Decorin' was not referenced in Fig 3a, while it was described in the figure legend. We apologize for this inadvertent omission, which is now corrected in the revised manuscript in the revised Fig. 3.

Figure 4

18) Fig 4b, did decorin treatment change the optical nerve volume? We did not observe any change in *Nfl* optic nerve volumes following decorin treatment (revised Fig. 5f).

Fig. 5f. No change in *Nfl*-OPG optic nerve volumes were observed following decorin treatment relative to control PBS-treated mice.

19) Fig 4c-e: Please provide the NFκB level/activity, e.g., p65 phosphorylation and nuclei translocation

and DNA binding. As recommended by the reviewer, we now include p65 phosphorylation and nuclear translocation in the revised Fig. 6d-f.

Fig. 6. (d) Decorin (800 pg/ml) blocks the increased p65-NFκB Ser⁵³⁶ phosphorylation induced by Ccl4 treatment ($n=3$). **(e)** Decorin (800 pg/ml) blocks the increased total p65-NFκB expression, **(f)** as well as the nuclear localization of p65-NFκB, induced by Ccl4 treatment ($n=3$). β-actin and HDAC are used as controls for total protein expression and the nuclear fractions, respectively.

20) Fig 4f, did NF-κB inhibitor change optical nerve volume? Yes, we observed a decrease in *Nfl* optic nerve volume following NFκB inhibition (revised Fig. 6h), likely reflecting the effects of this inhibitor on other cell types in the tumor ecosystem.

Fig. 6h. Schematic representation of the NFκB inhibitor treatment used. *Nfl*^{OPG} mice were treated between 4 and 6 weeks of age with the CAPE NFκB inhibitor ($n=8$), whereas control *Nfl*^{OPG} mice received PBS only ($n=8$). Isolated optic nerves were analyzed at 12 weeks of age. NFκB inhibitor treatment reduced optic glioma volume and proliferation (%Ki67⁺ cells), as well as microglia (%Iba1⁺ cells) and T cell (CD3⁺) content within the optic nerves of *Nfl*^{OPG} mice relative to vehicle-treated controls. Two-tailed Student's *t*-test.

21) Fig 4f, NFκB directly reduced the number of microglia, does it inhibit microglia expansion or reduce microglia viability? NFκB inhibition had no effect on microglia viability, which we now include in the (revised Extended Data Fig. 6e).

Extended Data Fig. 6e. NFκB inhibition (CAPE) had no effect on microglia viability, as measured using a WST-1 cell viability assay.

Reviewer 4

The article by Chatterjee et al describes a novel mechanism underlying the clinical observation that children with asthma have a reduced incidence of brain tumour, in particular optic gliomas. They identify a putative role for decorin, a small leucine rich proteoglycan, which deposition has been shown to be altered in asthma, and which deficiency in a mouse model leads to a mitigated response to allergen sensitization and challenge. The authors define a mechanistic pathway for the effect of decorin, implicating Cc15 production and NFkB inhibition, and leading to reduced glioma formation in a well characterized pre-clinical mouse model of low grade glioma. This is a very interesting study, crossing disciplines and opening up a new line of inquiry. The experimental approach is novel, well developed and has potential significance for an important clinical problem.

Major comments

1) Given the differences in the model, the authors should have more carefully described the “asthma-like” changes induced. They show representative images of bronchial walls, with evidence of thickening. This is not sufficient. We appreciate the comment by this reviewer. To provide more detailed analyses of the asthma induction, we worked closely with co-author, Dr. Michael Holtzman, an international authority on asthma to review the lung specimens. We now provide H&E and PAS staining of the asthma-like changes in wild type and *Nf1*^{+/-} mice, as well as measurements of serum IgG levels (revised Extended Data Fig. 1a- c).

Extended Data Fig. 1. WT and *Nf1*^{+/-} mice were treated between 4-6 weeks of age with either OVA or HDM (experimental asthma) or PBS (controls). Representative (a) H&E and (b) PAS-stained images of the lungs from WT and *Nf1*^{+/-} mice treated with PBS (*n*=6), HDM (*n*=6) and OVA (*n*=6). Increased alveolar wall thickening and immune cell infiltration were observed 7 days after OVA and HDM treatment. Boxes indicate thickened bronchial walls. Small scale bar, 500 μm; large scale bar, 250 μm. (c) Similar increases in serum IgE were observed in WT (*n*=4) and *Nf1*^{+/-} (*n*=4) mice treated with OVA and HDM relative to PBS-treated control mice. Data are presented as the mean ± SEM. One-way ANOVA with Bonferroni post-test correction; n.s., not significant; Exact P values are indicated within each panel. From left to right in each panel: (c) ns, ns, ns, P<0.0001.

2) The RNA sequencing results (Fig 3j) are very interesting and point to decorin as the molecule of interest. Biglycan is another SLRP with

similar/overlapping actions as decorin. It would be interesting to know if biglycan was altered in this model system. We thank the reviewer for this excellent suggestion. In contrast to decorin, biglycan had no effect on Ccl4 stimulation of microglia Ccl5 production (revised Extended Data Fig. 6b).

Extended Data Fig. 6b. WT microglia were treated with either biglycan (25 µg/ml) alone or in combination with Ccl4 (6000pg/ml). Biglycan alone or in combination with Ccl4 does not induce microglia Ccl5 production (n=5).

3) The published data on changes in decorin in human asthma are inconsistent. Huang et al (AJRCCM, 160:725,1999) showed no difference in decorin deposition in the airway walls of mild atopic asthmatics, as did Pini et al (AJP Lung: Cell Mol Physiol, 290:1052,2006) in the airway walls of moderate and severe asthmatics. On the other hand, de Medeiros Matsushita et al (J Pathol, 207:102, 2005) showed a decrease in decorin in the airway wall of patients dying with severe asthma. As such, the authors data (fig 3m) showing an increase in Dcn expression in CD4 and CD8 cells is not in keeping with that previously reported in the literature. This warrants some discussion. Can the authors provide more information about the patient population? (ref 55 reports on an animal model). Can the authors reconcile the lymphocyte data showing an increase in Dcn with the airway wall data showing stable or decreased decorin deposition? There were scant datasets available in GEO for an analysis of CD4 and CD8 T cells in human asthma. To this end, we specifically chose the GSE31773 dataset, since it contained data on both T cell populations in healthy adults and adults with non-steroid-dependent asthma⁸. The inclusion of patients without long-term steroid use eliminated the potential confounding effect on steroids on decorin expression, as reported by others in the literature^{9,10,11}. We did not examine decorin levels in the airways of our HDM/OVA-treated mice, only in the blood.

Minor comments

4) Did authors look at cytokines that are typically altered in asthma (IL4, IL5, IL3) to determine their potential impact on the chemokines of interest? As suggested by the reviewer, we performed gene expression analysis with cytokines typically altered in asthma (IL-3, IL-4, and IL-5) on the optic nerves from Nfl-OPG mice treated with PBS and OVA. We then treated T cell and microglia with the one cytokine increased in OVA-treated mice (IL-4), and did not see any change in Ccl4 and Ccl5 expression (revised Extended Data Fig. 4h-l).

Extended Data Fig. 4. (h-j) *Il3*, *Il4* and *Il5* gene expression was examined in PBS- and OVA-treated *Nfl*^{OPG} optic nerves (*n*=3), and only *Il3* expression was higher in OVA-treated *Nfl*^{OPG} mice. A two-tailed Student's *t*-test was used. **(k-l)** WT T cell (*n*=3) and microglia (*n*=3) were treated with IL-3 (1 ng/mL). No change in T cell *Ccl4* or microglia *Ccl5* production was observed following treatment with IL-3. A two-tailed Student's *t*-test was used.

5) Mice were administered decorin and the NFKB inhibitor by ip injection every other day for 2 weeks between 4 and 6 weeks of age. How did the authors choose this mode/dose of treatment? Is 1 mg/ml within the physiologic range? Were there off target effects? Why was the Dcn effect more specific than the NFKB effect (fig 4f)? We now provide a source for the decorin and NFkB treatments^{12,13}, as recommended. Since we could not measure decorin in the optic nerves following treatment, we cannot comment on the tissue levels relative to the OVA/HDM-treated T cell conditioned medium. We did not observe any obvious off-target effects in the limited treatment interval used. Finally, we hypothesize that the decorin effect might be more limited than the NFkB treatment, since decorin would act primarily on the microglia to dampen *Ccl5* production, while NFkB inhibits more than the *Ccl5* pathway in microglia and likely has other effects on microglia function.

6) Line 192: References 28 and 29 do not report data in asthma. We apologize for this inadvertent error, which is now corrected in the revised manuscript on page 21.

7) Ref 30 is the same as ref 17. We apologize for this inadvertent error, which is now corrected in the revised manuscript on page 21.

REFERENCES

1. Koga, M., Koga, K., Nakayama, J. & Imafuku, S. Anthropometric characteristics and comorbidities in Japanese patients with neurofibromatosis type 1: A single institutional case-control study. *J. Dermatol.* **41**, 885–889 (2014).
2. Mansouri, A. *et al.* Neurofibromatosis Clinic: A Report on Patient Demographics and Evaluation of the Clinic. *Can. J. Neurol. Sci.* **44**, 577–588 (2017).
3. Geller, M., Ribeiro, M. G., Araújo, A. P., De Oliveira, L. J. B. & Nunes, F. P. Serum IgE levels in neurofibromatosis 1. *Int. J. Immunogenet.* **33**, 111–115 (2006).
4. Bennett, M. L. *et al.* New tools for studying microglia in the mouse and human CNS. *Proc. Natl. Acad. Sci. U. S. A.* **113**, E1738–E1746 (2016).
5. Cichowski, K. *et al.* Mouse models of tumor development in neurofibromatosis type 1. *Science* (80-.). **286**, 2172–2176 (1999).
6. Tischler, A. S., Shih, T. S., Williams, B. O. & Jacks, T. Characterization of Pheochromocytomas in a Mouse Strain with a Targeted Disruptive Mutation of the Neurofibromatosis Gene *Nf1*. *Endocr. Pathol.* **6**, 323–335 (1995).
7. Jacks, T. *et al.* Tumour predisposition in mice heterozygous for a targeted mutation in *Nf1*. **7**, 353–361 (1994).
8. Jacks, T. *et al.* Tumour predisposition in mice heterozygous for a targeted mutation in *Nf1*. **7**, 353–361 (1994).
9. Kunz, L. I. Z. *et al.* Inhaled Steroids Modulate Extracellular Matrix Composition in Bronchial Biopsies of COPD Patients: A Randomized, Controlled Trial. *PLoS One* **8**, 6–13 (2013).
10. Wu, W. X., Zhang, Q., Unno, N., Derks, J. B. & Nathanielsz, P. W. Characterization of decorin mRNA in pregnant intrauterine tissues of the ewe and regulation by steroids. *Am. J. Physiol. - Cell Physiol.* **278**, 199–206 (2000).
11. Kahari, V. M., Lakkinen, L., Westermarck, J. & Larjava, H. Differential regulation of decorin and biglycan gene expression by dexamethasone and retinoic acid in cultured human skin fibroblasts. *J. Invest. Dermatol.* **104**, 503–508 (1995).
12. Border, W. A. *et al.* Natural inhibitor of transforming growth factor- β protects against scarring in experimental kidney disease. *Nature* **360**, 361–364 (1992).
13. Morroni, F. *et al.* Neuroprotective effect of caffeic acid phenethyl ester in a mouse model of alzheimer's disease involves Nrf2/HO-1 pathway. *Aging Dis.* **9**, 605–622 (2018).

Reviewers' Comments:

Reviewer #2:

Remarks to the Author:

The authors did a very thorough job in addressing my concerns / questions.

Reviewer #3:

Remarks to the Author:

The authors have thoroughly revised the manuscript and have addressed the concerns raised in the earlier round of review. Very nice paper, congratulations!

Reviewer #4:

Remarks to the Author:

The additional experiments and responses to reviewer have addressed my concerns. My only minor comment is that ref 21 and 34 reference the same manuscript.

Reviewer #2

1. The authors did a very thorough job in addressing my concerns / questions. *We thank the reviewer for their positive comment.*

Reviewer #3

1) The authors have thoroughly revised the manuscript and have addressed the concerns raised in the earlier round of review. Very nice paper, congratulations! *We thank the reviewer for their positive appraisal.*

Reviewer 4

The additional experiments and responses to reviewer have addressed my concerns. My only minor comment is that ref 21 and 34 reference the same manuscript. *We thank the reviewer for their positive comments. We fixed the references noted.*

In addition, we made the requested editorial changes suggested, and completed the reporting summary. We hope that these changes now make our manuscript acceptable for publication in *Nature Communications*.